# Most Activation Functions Can Win the Lottery Without Excessive Depth

**Rebekka Burholz**
CISPA Helmholtz Center for Information Security
66123 Saarbrücken, Germany
`burkholz@cispa.de`

## Abstract

The strong lottery ticket hypothesis has highlighted the potential for training deep neural networks by pruning, which has inspired interesting practical and theoretical insights into how neural networks can represent functions. For networks with ReLU activation functions, it has been proven that a target network with depth $L$ can be approximated by the subnetwork of a randomly initialized neural network that has double the target's depth $2L$ and is wider by a logarithmic factor. We show that a depth $L + 1$ network is sufficient. This result indicates that we can expect to find lottery tickets at realistic, commonly used depths while only requiring logarithmic overparametrization. Our novel construction approach applies to a large class of activation functions and is not limited to ReLUs. Code is available on Github ([RelationalML/LT-existence](#)).

## 1 Introduction

The Lottery Ticket Hypothesis [13] and, in particular, its strong version [38] postulate that pruning deep neural networks might provide a promising alternative to training large, overparameterized neural networks with Stochastic Gradient Descent (SGD). Pruning has the potential to not only identify small scale neural networks that possess a meaningful, task-specific neural network structure and generalize better due to regularization [51], but also to reduce the computational burden associated with deep learning [50, 10, 29].

Also from a theoretical perspective, it has been shown that for every small enough target network a sufficiently large, randomly initialized neural network, the source network, contains a subnetwork, the lottery ticket (LT), that can approximate the target up to acceptable accuracy with high probability. [32] has been the first to provide a probabilistic lower bound on the required network width of the larger random network. This bound has been improved by [37, 36] to a width that is larger than a target's width only by a logarithmic factor. The limitations of these works are that they all are restricted to ReLU activation functions and assume that the larger random network has twice the depth of the target network $L_s = 2L_t$.

However, it is well known that deeper networks tend to have a higher expressiveness, as they can approximate certain function classes with significantly fewer parameters than their shallow counterparts [33, 49]. It could therefore reduce the overall sparsity of the target and the LT to allow the target to utilize more of the source depth $L_s$ for a sparser representation. A lower depth requirment could also [11] have therefore derived lower bounds for the width of the random network of depth $L_s = L_t + 1$, but these can only cover extremely sparse target networks, as the width requirement is polynomial in the inverse approximation error $1/\epsilon$.

In contrast, we derive a novel construction that only requires a logarithmic factor. While we utilize subset sum approximation results like [37], we allow a target network to have almost the same depth

as the source network with depth $L_s \geq L_t + 1$ instead of $L_s = 2L_t$. Our derivations further apply to a large class of activation functions that includes but is not limited to ReLUs.

The reduced depth requirement and flexibility in the activation function of our construction can result in significantly sparser neural network target networks and thus also LTs. For example, $f(x) = x^2$ can be approximated up to error $\epsilon$ by a shallow ReLU network with $O(1/\epsilon)$ parameters, while a deep enough network only needs $O(\log(1/\epsilon))$ parameters [33]. Also the activation function determines the possible target sparsity, e.g., $x^2$ can be represented with only 4 parameters for all $\epsilon > 0$ if $\phi(x) = (\max\{x, 0\})^2$. Therefore, the potential sparsity of the final lottery ticket depends critically on the choice of activation functions and architecture of the source network, including its depth and width. We provide a theoretical foundation that provides flexibility regarding these choices.

**Contributions** 1) We prove the existence of strong lottery tickets as subnetworks of a larger randomly initialized source neural network for a large class of activation functions, including ReLUs. 2) We derive a novel construction that requires the source network to have almost the same depth $(L + 1)$ as the target network, thus, allowing it to leverage the potential representational benefits associated with larger depth. 3) Despite the reduced depth requirement we keep the width requirement logarithmic in the approximation error $\epsilon$. 4) Our proofs are constructive and define an algorithm that approximates a given target network by pruning a source network. We verify in experiments that this algorithm is successful under realistic conditions.

**Related Literature** Many pruning methods have been proposed to reduce the number of neural network parameters during training [22, 35, 19, 13, 43, 24, 50, 14, 39, 28, 27, 48, 40, 6] or thereafter [41, 23, 20, 9, 26, 34, 52] and have been applied in different contexts, including graph neural networks [5] and GANs [4]. Furthermore, pruning can have provable regularization and generalization properties [51]. The algorithms are most useful for structure learning at lower sparsity levels [44, 24] but at least Iterative Magnitude Pruning (IMP) [19, 13] can fail in identifying LTs that perform superior to random or smaller dense networks [31]. It can therefore be beneficial to start pruning from a sparse random architecture rather than a dense network [10, 29], which saves computational resources. Other options to achieve the latter are to identify and train on core sets [53] and focus on pruning before training [47, 25, 46, 45, 38]. Yet, iterative pruning methods often perform better [15, 31] but all face challenges in finding highly sparse LTs [11].

Most of the discussed pruning methods try to find LTs in a 'weak' (but powerful) sense by identifying a sparse neural network architecture that is well trainable starting from its initial parameters. Strong LTs are sparse subnetworks that perform well with the initial parameters and, hence, do not need further training [54, 38]. Their existence has been proven for fully-connected feed forward networks with RELU activation functions by providing lower bounds on the width of the large, randomly initialized source network [32, 37, 36, 12, 3, 17]. In addition, it was shown that multiple candidate tickets exist that are also robust to parameter quantization [8]. Our first objective is to extend the known theory to other activation functions beyond RELUs. Note that the approach that we develop here has also been partially transferred to convolutional and residual architectures [1]. However, [1] does not cover activation functions with nonzero intercept (like sigmoids) and their width dependence on the error $\epsilon$ is less advantageous.

## 2   Constructing Lottery Tickets

Informally, our goal is to show that any deep neural network of width $n_t$ and depth $L_t$ or smaller can be approximated with probability $1 - \delta$ up to error $\epsilon$ by pruning a larger randomly initialized neural network of width $O(n_t \log[n_t L_t / \min\{\delta, \epsilon\}])$ and depth $2L_t$ or a network of smaller depth $L_t + 1$ and width $O(n_t \log[n_t L_t \log(1/\delta) / \min\{\delta, \epsilon\}])$, as long as the target network and the large random network rely on the same activation functions. Note that we often omit the dependence on $\delta$ in our discussions because we usually care about cases when $\epsilon < \delta$. Next we explain the required background to formalize and prove our claims.

**Background and Notation** Let a fully-connected feed forward neural network $f : \mathcal{D} \subset \mathbb{R}^{n_0} \to \mathbb{R}^{n_L}$ be defined on a compact domain $\mathcal{D}$ and have architecture $\bar{n} = [n_0, n_1, ..., n_L]$, i.e., depth $L$ and width $n_l$ in layer $l \in (0, ..., L)$, with continuous activation function $\phi(x)$. On the relevant compact domain let $\phi(x)$ have Lipschitz constant $T$. It maps an input vector $\boldsymbol{x}^{(0)}$ to neurons $x_i^{(l)}$ as

$\boldsymbol{x}^{(l)} = \phi\left(\boldsymbol{h}^{(l)}\right)$ with $\boldsymbol{h}^{(l)} = \boldsymbol{W}^{(l)}\boldsymbol{x}^{(l-1)} + \boldsymbol{b}^{(l)}$, where $\boldsymbol{h}^{(l)}$ is the pre-activation, $\boldsymbol{W}^{(l)} \in \mathbb{R}^{n_l \times n_{l-1}}$ is the weight matrix, and $\boldsymbol{b}^{(l)} \in \mathbb{R}^{n_l}$ is the bias vector of layer $l$. Without loss of generality, let us assume that each parameter (weight or bias) $\theta$ is bounded as $|\theta| \leq 1$. Primarily, we distinguish three networks. First, we want to approximate a target network $f_t$ with architecture $\bar{n}_t$ of depth $L_t$ consisting of $N_t$ total nonzero parameters. Second, this approximation is performed by a lottery ticket (LT) $f_\epsilon$ that is obtained by pruning a larger source network $f_s$, which we indicate by writing $f_\epsilon \subset f_s$. Third, this source network is a larger fully-connected feed-forward, randomly initialized neural network with architecture $\bar{n}_s$ and depth $L_s$. While most LT existence results require exactly $L_s = 2L_t$, we show that any $L_s \geq L_t + 1$ is sufficient.

To simplify the presentation, like most works, we assume a convenient parameter initialization that we have to choose with respect to the activation function if we approximate a target layer with two source network layers. In most cases, we make the following assumption.

**Assumption 2.1** (Convenient initialization). We assume that the parameters of the source network $f_s$ are independently distributed as $w_{ij}^{(l)} \sim U\left([-1,1]\right)$, $b_i^{(1)} \sim U\left([-1,1]\right)$ and $b_i^{(l)} = 0$ for $l > 1$.

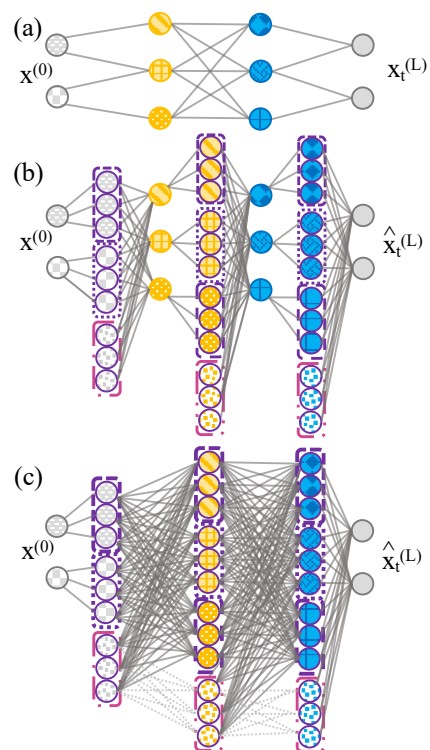

(a)

$\mathbf{x}^{(0)}$ $\qquad$ $\mathbf{x}_t^{(L)}$

(b)

$\mathbf{x}^{(0)}$ $\qquad$ $\hat{\mathbf{x}}_t^{(L)}$

(c)

$\mathbf{x}^{(0)}$ $\qquad$ $\hat{\mathbf{x}}_t^{(L)}$

Note that also other parameter distributions, e.g. normal distributions, are covered as long as they contain a uniform distribution [37], since this allows us to solve subset sum approximation problems (see appendix for a formal definition). While most works assume zero biases ($b_i^{(l)} = 0$), as they focus on target networks without biases [32, 37], we initialize the biases in the first layer as nonzero. This is sufficient to approximate nonzero target biases, since we can always construct constant neurons in each layer (see pink blocks in Fig. 1 (c)).

The convenient parameter initialization can always be transferred to a realistic one by learning or just applying an appropriate layer-specific scaling factor $\lambda_l$, as also proposed in LT pruning experiments [54]. For homogeneous activation functions like RELUs, even a global parameter scaling is sufficient [12]. For instance, let us assume that our initial parameters in Layer $l$ are distributed as $U\left([-\sigma_l, \sigma_l]\right)$. A common choice would be a He [21] or Glorot [18] initialization with $\sigma_l \propto 1/\sqrt{n_l}$. In this case, we would need to adapt our proofs by replacing $\{\theta^l\}$ of a LT by $\{\lambda_l\theta^l\}$ with $\lambda_l = 1/\sigma_l$ to construct the same function that we have derived for convenient initializations.

Figure 1: LT construction idea. (a) Target network $f_t$. (b) $L_0 = 2L_t$ construction of $f_\epsilon$. (c) $L_0 = L_t + 1$ construction of $f_\epsilon$. Subset sum blocks are framed (purple corresponding to target neurons, pink to biases). Dashed links only exist if source network biases in layers $l > 1$ are initialized to zero.

For specific activation functions with $\phi(0) \neq 0$ (e.g. SIGMOIDS), instead of Assumption 2.1, we will assume an initialization that has originally been derived to ensure dynamical isometry for RELUs [2, 16]. Interestingly, this initialization also supports LTs.

**Assumption 2.2** (Looks-linear initialization). We assume that the weight matrices of the source network $f_s$ are initialized as $W^{(l)} = \begin{bmatrix} M^{(l)} & -M^{(l)} \\ -M^{(l)} & M^{(l)} \end{bmatrix}$, where $M^{(l)}$ and $b^{(l)}$ are distributed as in Assumptions 2.1.

In general, we measure the approximation error with respect to the supremum norm, which is defined as $\|g\|_\infty := \sup_{\mathbf{x} \in \mathcal{D}} \|g\|_1$ for any function $g$ and the L1-norm $\|x\|_1 := \sum_i |x_i|$. In the formulation of theorems, we also make use of the universal constant $C$ that can attain different values.

**Lottery Tickets as Subnetworks** [32] were the first to give probabilistic guarantees and provide a lower bound on the required width of the source network that is polynomial in the width of the target

network. Their $O(n_t^5 L_t^2/\epsilon^2)$ requirement, or under additional sparsity assumptions $O(n_t^2 L_t^2/\epsilon^2)$, has been improved to a logarithmic dependency of the form $O(n_t^2 \log(n_t L_t)/\epsilon)$ for weights that follow an unusual hyperbolic distribution [36] and $O(n_t \log(n_t L_t/\epsilon))$ for uniformly distributed weights [37]. While these assume target networks with zero biases, [12] transferred the approach by [37] to nonzero biases. All assume that the source network has exactly twice the depth of the target network ($L_s = 2L_t$). Only [11] prove existence for extremely sparse tickets for $L_s = L_t + 1$ but the width requirements are unrealistic for most target architectures. [3] show how to leverage additional depth $L_s \geq 2L_t$ but still assume excessive depth in general. Furthermore, all of these works focus on RELU activation functions. Both limitations, the focus on RELUS and the $L_s = 2L_t$ requirement rely on the following construction idea that is also visualized in Fig. 1(a-b). Every layer of the target network $f_t$ is represented by two layers in the lottery ticket $f_\epsilon$ and equipped with RELU activation functions $\phi_R(x) = \max\{x, 0\}$ with the possible exception of the last output layer. We can obtain two layers by representing the identity as $x = \phi_R(x) - \phi_R(-x)$ and writing each target neuron $x_{t,i}^{(l)}$ as $x_{t,i}^{(l)} = \phi_R\left(\sum_j w_{t,ij}^{(l)} x_{t,j}^{(l-1)} + b_{t,i}^{(l)}\right) = \phi_R\left[\sum_j w_{t,ij}^{(l)}\phi_R\left(x_{t,j}^{(l-1)}\right) - w_{t,ij}^{(l)}\phi_R\left(-x_{t,j}^{(l-1)}\right) + b_{t,i}^{(l)}\right]$.
The advantage of this 2-layer representation is that (a) we gain the flexibility to select the neurons in the middle layer among a higher number of available neurons in the source network and (b) these neurons are univariate so that they depend on a single nonzero parameter and are thus simple to approximate with high probability. The question is how many nodes $n_0$ in the middle layer of the source network are required to guarantee an accurate selection. The answer distinguishes previous work. [37] achieve a factor $\gamma$ (where $n_s = \gamma n_t$) that is logarithmic in $n_s$, $\epsilon$, $\delta$, etc. by solving a separate subset sum approximation problem for each parameter utilizing results by [30]. [12] transfers these results to target networks with nonzero biases.

Our first contribution is to extend a similar construction to a wider class of activation functions that is not restricted to RELUS. We need this result to approximate at least the first layer of our target network in our second contribution, an $L + 1$-construction, as shown in Fig. 1(c). For the remaining layers, we propose to construct the subset sum blocks for the next layer directly from the previous layer by sharing nodes in the construction. This has multiple advantages. The obvious one is that we can use the available depth of the source network to start from a potentially sparser target architecture to solve a given problem. Also the LT itself consists usually of less neurons. Furthermore, the subset sum approximation of parameters becomes more efficient.

**Subset Sum Approximation**    In the discussed construction, we generally have multiple random neurons and parameters available to approximate a target parameter $z$ by $\widehat{z}$ up to error $\epsilon$ so that $|z - \widehat{z}| \leq \epsilon$. Let us denote the independent random variables that we can use for this approximation as $X_1, ..., X_m$. If they contain uniform distributions, Lueker [30] has shown that $m \geq C \log(1/\min\{\epsilon, \delta\})$ is sufficient for the existence of a subset $I \subset \{1..., m\}$ so that the approximation of $z$ by $\widehat{z} = \sum_{k \in I} X_k$ is successful with probability at least $1 - \delta$. Standard distributions of interest like uniform and normal distributions as well as their products have this property [37]. For convenience, a precise corollary is stated in the appendix as Cor. A.2. Note that $C$ depends on the distributions of the $X_k$ and is usually larger in the two-layers-for-one than in the one-layer-for-one construction, because the $X_k$ are given by products $X_k = w_{0,ik}^{(l+1)} w_{0,kj}^{(l)}$ in the former but $X_k = w_{0,ik}^{(l+1)}$ in the latter case.

## 2.1    Two Layers for One

Our first contribution is to transfer the two-layers-for-one construction to activation functions that can be different from ReLUs. We will utilize this result also in our one-layer-for-one construction to represent the first layer of the target network. This is necessary as the input neurons are fixed. We need to increase their multiplicity in the first layer of the source network to solve subset sum approximation problems that create target Layer 1 in the LT's Layer 2.

### 2.1.1    Activation Functions

The main property of ReLUs $\phi_R$ that is utilized in LT existence proofs is that we can represent the identity as $x = \phi_R(x) - \phi_R(-x)$. This identity is exact and holds for all inputs $x \in \mathbb{R}$. Yet, we can show that, in combination with the right initialization of the source network, it is sufficient if we can approximate the identity on an interval that contains 0. This approximation is feasible with most continuous activation functions that are not constant zero in a neighborhood of 0, as we detail next.

**Assumption 2.3** (Activation function (first layer)). For any given $\epsilon' > 0$ exists a neighborhood $[-a(\epsilon'), a(\epsilon')]$ of 0 with $a(\epsilon') > 0$ so that the activation function $\phi$ can be approximated by $\widehat{\phi}(x)$ on that neighborhood such that $\sup_{x \in [-a,a]} |\phi(x) - \widehat{\phi}(x)| \leq \epsilon'$, where $\widehat{\phi}(x) = m_+ x + d$ for $x \geq 0$ and $\widehat{\phi}(x) = m_- x + d$ for $x < 0$ with $m_+, m_-, d \in \mathbb{R}$, and $m_+ + m_- \neq 0$. We further assume that, if $a(x)$ is finite, $g(x) = x/a(x)$ is invertible on an interval $]0, \epsilon'']$ with $\epsilon'' > 0$ and $\lim_{x \to 0} g(x) = 0$.

Most continuous functions and thus most popular activation functions fulfill this assumption. For instance, RELUS $\phi_R(x) = \max(x, 0)$ inflict zero error on $\mathbb{R}$ (i.e., $a = \infty$) with $m_+ = 1$, $m_- = 0$, and $d = 0$. Similarly, LRELUS can be represented without error by $m_+ = 1$, $m_- = \alpha$, and $d = 0$ for an $\alpha > 0$. $\phi(x) = \tanh(x)$ is approximately linear so that $|\tanh(x) - x| \leq x^3/3$ for $|x| < \pi/2$, which can be seen by Taylor expansion of $\tanh$. This implies that the choice $m_+ = 1$, $m_- = 1$, and $d = 0$ with $a = \min\{(3\epsilon')^{1/3}, \pi/2\}$ fulfills our assumption. SIGMOIDS $\phi(x) = 1/(1 + \exp(-x))$ can be analyzed in the same way with $m_+ = m_- = 0.25$, $d = 0.5$, and $a = \min\{(48\epsilon')^{1/3}, \pi\}$, since $\phi(x) = (\tanh(x/2) + 1)/2$. We should point out that not all continuous functions fulfill this property. Counterexamples include a shifted ReLU $\phi(x) = \phi_R(x - 1)$ or $\phi(x) = |x|$. However, in our $L + 1$-construction, almost all activation functions can be arbitrary continuous functions. We will only ask the activation functions in the first layer to meet our assumption above. How can we represent the identity with such activation functions?

**Lemma 2.4** (Representation of the identity). *For any $\epsilon' > 0$, for a function $\phi(x)$ that fulfills Assumption 2.3 with $a = a(\epsilon') > 0$, and for every $x \in [-a, a]$ we have*

$$\left| x - \frac{1}{m_+ + m_-} \left( \phi(x) - \phi(-x) \right) \right| \leq \frac{2\epsilon'}{m_+ + m_-}. \tag{1}$$

Note that RELUS and LRELUS inflict no approximation error so that the above statement holds also for $\epsilon' = 0$ and $a = \infty$. Some activation functions can have other advantages over RELUS. For instance, functions for which $m_+ = m_- = m$ and $d = 0$ like TANH can also be approximated by $|x - \phi(x)/m| \leq \epsilon'/m$ and do not need separate approximations of the positive and the negative part. Furthermore, SIGMOIDS and general activation functions with $d \neq 0$ do not need nonzero biases in the source network, since we can approximate a bias by random variables of the form $X_k = w_k \phi(0)$.

The challenge in utilizing this lemma is to incorporate the error that is inflicted by the approximation of the identity in addition to the pruning error into the analysis of the total approximation error. The allowed interval size $a$ also influences which variables can enter our subset sum blocks and could affect our width requirement if we would not initialize our parameters in the right way.

### 2.1.2 Lottery Ticket Existence (Two-for-One)

Our first goal is to identify a LT $f_\epsilon$ that approximates a single hidden layer neural network $f_t(x) : \mathcal{D} \subset \mathbb{R}^{n_0} \to \mathbb{R}$ with $f_t(x) = \phi_t \left( \sum_{j=1}^{n_0} w_{t,j} x_j + b_t \right)$. We could easily extend this result to approximate each layer of a multilayer neural network but we will discuss a more promising alternative that uses the following result for approximating the first layer only. $f_\epsilon$ can be obtained by pruning a fully-connected source network $f_s$ of depth $L_s = 2$ with $n_{s,1}$ hidden neurons that are equipped with the activation function $\phi_0$, which can be different from the outer $\phi_t$. The additional layer in $f_s$ has the purpose to create multiple copies of input nodes that can be used in subset sum approximations.

To achieve this despite the potential non-linear activation function $\phi_0$ in the hidden layer, we have to approximate the identity with the help of $\phi_0$. The precise approximation can pose two additional challenges in comparison with the standard construction for RELUS. (a) If the piece-wise linear approximation of $\phi_0$ in Assumption 2.3 holds only on a finite neighborhood of 0 with $a(\epsilon) < \infty$, we can only use parameters that render the approximation valid. Thus, the success of our approach relies on an appropriate parameter initialization in the first layer of the source network $f_s$. (b) A nonzero intercept ($d \neq 0$) in the approximation of $\phi_0$ demands for a different initialization scheme to avoid the need for large bias approximations. For simplicity, let us first assume that $\phi_0$ fulfills Assumption 2.3 with zero intercept ($d = 0$).

**Theorem 2.5** (LT Existence (Two-for-One)). *Assume that $\epsilon', \delta' \in (0, 1)$, a target network $f_t(x) : \mathcal{D} \subset \mathbb{R}^{n_0} \to \mathbb{R}^{n_1}$ with $f_{t,i}(x) = \phi_t \left( \sum_{j=1}^{n_0} w_{t,ij} x_j + b_{t,i} \right)$, and two-layer source network $f_s$ with architecture $[n_0, n_{s,1}, n_1]$ and activation functions $\phi_0$ in the first and $\phi_t$ in the second layer*

*are given. Let $\phi_0$ fulfill Assumption 2.3 with $a(\epsilon'') > 0$ and $d = 0$, $\phi_t$ have Lipschitz constant $T_t$, and $M := \max\{1, \max_{\mathbf{x} \in \mathcal{D}, i} |x_i|\}$. Let the parameters of $f_s$ be conveniently initialized as $w_{ij}^{(1)}, b_i^{(1)} \sim U[-\sigma, \sigma]$ and $w_{ij}^{(2)} \sim U[-1/(|m_+ + m_-|\sigma), 1/(|m_+ + m_-|\sigma)]$, $b_i^{(2)} = 0$, where $\sigma = \min\{1, a(\epsilon'')/M\}$ with $\epsilon'' = g^{-1}\left(\epsilon'/\left(CT_t n_0 \frac{M}{|m_+ + m_-|} \log\left(\frac{n_0}{\min\left\{\frac{\delta'}{n_{t,1}}, \frac{\epsilon'}{T_t M}\right\}}\right)\right)\right)$. Then, with probability at least $1 - \delta'$, $f_s$ contains a subnetwork $f_{\epsilon'} \subset f_s$ so that each output component $i$ is approximated as $\max_{\mathbf{x} \in \mathcal{D}} |f_{t,i}(\mathbf{x}) - f_{\epsilon',i}(\mathbf{x})| \leq \epsilon'$ if*

$$n_{s,1} \geq C n_0 \log\left(\frac{n_0}{\min\{\epsilon'/(T_t M), \delta'/n_{t,1}\}}\right).$$

*Proof Outline.* The construction of a LT consists of three steps. First, we prune the hidden neurons of $f_s$ to become univariate. Second, we identify neurons in the hidden layer for which we can approximate $\phi_0$ for small inputs according to Assumption 2.3. Third, if $n_{s,1}$ is large enough, we can select subsets $I_j$ and $I_b$ of the hidden neurons with small inputs so that we can use Cor. A.2 on subset sum approximation to approximate the parameters of the target. The resulting subnetwork of $f_s$ is of the following form $f_{\epsilon',i}(x) = \phi_t\left(\sum_{k \in I} w_{ik}^{(2)} \phi_0\left(w_{kj}^{(1)} x_j\right) + \sum_{k \in I_b} w_{ik}^{(2)} \phi_0\left(b_k^{(1)}\right)\right)$, where $I = \cup_j I_j$ and $\sum_{k \in I_j} w_{ik}^{(2)} w_{kj}^{(1)}$ can be used to approximate $w_{t,ij}$. $f_{\epsilon,i}$ qualifies as LT if $|f_{t,i}(x) - f_{\epsilon',i}(x)| \leq \epsilon'$. A straight-forward series of bounds shows that we can achieve this if $\Delta_j = \left|w_{t,ij} x_j - \sum_{k \in I_j} w_{ik}^{(2)} \phi_0\left(w_{kj}^{(1)} x_j\right)\right| \leq \epsilon'/(T_t(n_0 + 1))$ for each $j$. This also applies to the approximation of $b_{t,i}$, respectively. We would like to approximate $\phi_0\left(w_{kj}^{(1)} x_j\right) \approx \mu_\pm(w_{kj}^{(1)} x_j) w_{kj}^{(1)} x_j \pm \epsilon''$, where $\mu_\pm(x) = m_+ x$ if $x \geq 0$ and $\mu_\pm(x) = m_- x$ otherwise. Note that $\mu_\pm(x) + \mu_\pm(-x) = m_+ + m_-$ for all $x \neq 0$. By construction, the activation function approximation is valid, as $|w_{kj}^{(1)} x_j| \leq M a(\epsilon'')/M = a(\epsilon'')$. Hence, we can split the error into two subset sum approximation problems and the activation function approximation error: $\Delta_j \leq M \frac{|\mu_\pm(x_j)|}{|m_+ + m_-|} |w_{t,ij} - |m_+ + m_-| \sum_{k \in I_j^+} w_{ik}^{(2)} w_{kj}^{(1)}| + M\left(1 - \frac{|\mu_\pm(x_j)|}{|m_+ + m_-|}\right) |w_{t,ij} - |m_+ + m_-| \sum_{k \in I_j^-} w_{ik}^{(2)} w_{kj}^{(1)}| + |\sum_{k \in I_j} w_{ik}^{(2)}\left(\phi_0(w_{kj}^{(1)} x_j) - \mu_\pm(w_{kj}^{(1)} x_j) w_{kj}^{(1)} x_j\right)|$, where $I_j$ is split into indices $I_j^+$ for which $w_{kj}^{(1)} > 0$ and $I_j^-$ all the ones for which $w_{kj}^{(1)} < 0$. The first two terms can be bounded according to Cor. A.2 with probability $1 - \delta''$ if $C \log\left(\frac{1}{\min\{\epsilon'/(2(n_0+1)MT_t), \delta''\}}\right)$ random variables $X_k$ are available to choose from $I_j^\pm$, since the random variables $X_k = |m_+ + m_-| w_{ik}^{(2)} w_{kj}^{(1)}$ are distributed as $X_k \sim U[-1, 1]U[0, 1]$ or $X_k \sim U[-1, 1]U[-1, 0]$ and thus contain a uniform distribution $U[-1, 1]$ as shown by [37]. We can solve $2(n_0 + 1)n_{t,1}$ independent subset sum approximation problems with probability $1 - \delta'$ if each problem is solved with $\delta'' = \delta'/(2(n_0 + 1)n_{t,1})$ and if we have in total $C n_0 \log\left(\frac{n_0}{\min\{\epsilon'/(MT_t), \delta'/n_{t,1}\}}\right)$ random variables available. $\square$

The full proof is given in the appendix. A key result of our construction is that the activation function approximation error does not affect the width requirement. As long as we choose the scaling factor $\sigma$ small enough, we can achieve a small enough error. Note that for RELUs, the above theorem reduces to known results [12], since $\sigma = 1$, $m_+ + m_- = 1$, $a = \infty$, and $d = 0$. LRELUs have the same advantageous property and are now covered in addition. The only difference is that $m_+ + m_- = 1 + \alpha$. Another new insight is that also activation functions with finite approximation support $a(\epsilon'') < \infty$ support the existence of lottery tickets with the right parameter initialization with $0 < \sigma < 1$. Interestingly, they can still achieve the same realistic width requirement.

Even though the theorem does not provide details on the universal constant, the minimum width requirement in this construction would be achieved by a linear activation function $\phi_0(x) = x$, which does not require a distinction between positive and negative $w_{kj}^{(1)}$, has $a(\epsilon'') = \infty$, does not inflict any approximation error, and is homogeneous, which makes it easy to transfer the above results to realistic parameter initialization schemes.

Remember that only $\phi_0$ in the first layer needs to fulfill Ass. 2.3. $\phi_t$ is an arbitrary continuous function. However, it is relatively uncommon in practice to combine different activation functions in

the same neural network so that $\phi_0 \neq \phi_t$. Regardless, RELU neural networks have been observed to learn the identity in the first layers (close to the input) [42]. For these reasons it could be beneficial in general to equip at least the first layer with linear activation functions.

Also a 'looks-linear' parameter initialization can be of great benefit, in particular, if the intermediary activation function $\phi_0$ has a nonzero intercept $d \neq 0$, as shown next.

**Theorem 2.6** (LT Existence (Two-for-One) with Nonzero Intercept). *Thm. 2.5 applies also to activation functions $\phi_0$ that fulfill Assumption 2.3 with $d \neq 0$ if the parameters are initialized according to Assumption 2.2 with $M_0^{(l)}$ distributed as the weights in Thm. 2.5.*

*Proof Outline.* We can closely follow the steps of the previous proof. The major difference is that we approximate $\phi_0(x) \approx \mu_\pm(x)x + d$ so that the activation function approximation produces an additional error term, i.e., $\left| d \sum_{k \in I_j} w_{ik}^{(2)} \right|$. In principle, we could have modified the bias subset sum approximation by approximating $b_{t,i} + d \sum_j \sum_{k \in I_j} w_{ik}^{(2)}$ instead of $b_{t,i}$. Yet, $\sum_j \sum_{k \in I_j} w_{ik}^{(2)}$ could be a large number, with which we would need to multiply our width requirement, if each $w_{ik}^{(2)}$ is initialized as in Thm. 2.5. In contrast, with the looks-linear initialization we can choose $w_{ik'}^{(2)} = -w_{ik}^{(2)}$ so that $\sum_{k \in I_j} w_{ik}^{(2)} = \sum_{k \in I_j^+} w_{ik}^{(2)} + \sum_{k' \in I_j^-} w_{ik'}^{(2)} = \sum_{k \in I_j^+} w_{ik}^{(2)} - \sum_{k \in I_j^+} w_{ik}^{(2)} = 0$. □

As for RELUs [12], we could use these results to approximate every layer of a target multi-layer feed-forward neural network for general activation functions by pruning two layers of a source network with double the depth as the target network, as visualized in Fig. 1 (b). As alternative, next we propose to prune a source network that has almost the same depth as the target network.

## 2.2 One Layer for One

Our second major contribution is to show how we can approximate intermediate target layers by pruning a single layer of the source network. This is achieved by connecting subset sum approximation blocks directly as visualized in Fig. 1 (c). The main idea is to create, instead of a single target neuron, $\rho$ copies to support subset sum approximations in the next layer.

In comparison with the two-layer-for-one construction, we have to solve a higher number of subset sum approximation problems but this number is only higher by a logarithmic factor and can be integrated in the universal constant $C$. It is usually negligible. In total, the lottery ticket might also consist of a higher number of parameters, i.e., link weights, but a smaller number of neurons, which is the more critical case to reduce computational costs [34]. The benefits usually outweigh the costs, as we need less random variables to guaranty a successful subset sum approximation and, most importantly, can use almost the full depth of the source network to find a sparser representation of the target network. Furthermore, source networks of lower depth are easier to train and thus also more amenable to pruning algorithms that utilize gradient based algorithms.

Interestingly, we can further relax our requirements on the activation functions, as they only influence the error propagation through the network but are not involved anymore in creating random variables for the subset sum approximation problems. Let us start with the approximation of a single layer.

**Theorem 2.7.** *Assume that $\epsilon', \delta' \in (0, 1)$, a target network layer $f_t(x) : \mathcal{D} \subset \mathbb{R}^{n_{t,l}} \to \mathbb{R}^{n_{t,l+1}}$ with $f_{t,i}(x) = \phi_t \left( \sum_{j=1}^{n_{t,l}} w_{t,ij} x_j + b_{t,i} \right)$, and one source network layer $f_s$ with architecture $[n_{s,l+1}, n_{s,l+2}]$ and the same activation function $\phi_t$ are given. Let $\phi_t$ have Lipschitz constant $T$, and define $M := \max\{1, \max_{\mathbf{x} \in \mathcal{D}, i} |x_i|\}$. Let the parameters of $f_s$ be initialized according to Assumption 2.1. Then, with probability at least $1 - \delta'$, $f_s$ contains a subnetwork $f_{\epsilon'} \subset f_s$ so that $\rho$ copies of each output component $i$ are approximated as $\max_{\mathbf{x} \in \mathcal{D}} |f_{t,i}(\mathbf{x}) - f_{\epsilon',i'}(\mathbf{x})| \leq \epsilon'$ if*

$$n_{s,l+1} \geq C n_{t,l} \log \left( \frac{n_{t,l}}{\min\{\epsilon'/(MT), \delta'/(\rho n_{t,l+1})\}} \right).$$

*Proof.* For each target input neuron $j$, we assume that we can create multiple copies (with index set $I_j$) that we can use then as basis for subset sum approximations of target weights $w_{t,ij}^{(l+1)}$. Similarly, for biases we reserve neurons $I_b$ in the first layer that are pruned so that we have $\phi_t(c) =$

1 or another constant. Particularly parameter efficient would be $\phi_t(0) = d$. Thus, our lottery ticket is of the form $f_{\epsilon',i'}(x) = \phi_t\Big(\sum_{j=1}^{n_{t,l}} x_j \sum_{k \in I_{i'j} \subset I_j} w_{i'k}^{(l+2)} + \sum_{k \in I_{i'b} \subset I_b} w_{i'k}^{(l+2)} \phi_t(c)\Big)$. The subsets $I_{i'j}$ and $I_{i'b}$ are chosen so that $|w_{t,ij}^{(l+1)} - \sum_{k \in I_{i'j} \subset I_j} w_{i'k}^{(l+2)}| \leq \epsilon'/(MT(n_{t,l}+1))$ and $|b_{t,i}^{(l+1)} - \sum_{k \in I_{i'b} \subset I_b} w_{i'k}^{(l+2)} \phi_t(b_k^{(l+1)})| \leq \epsilon'/T(n_{t,l}+1)$, which can be achieved by subset sum approximation based on the random variables $X_k = w_{i'k}^{(l+2)} \sim U[-1,1]$. In total, we have to solve maximally $\rho n_{t,l+1}(n_{t,l}+1)$ of these problems and can thus spend $\delta'/(\rho n_{t,l+1}(n_{t,l}+1))$ on each of them. Note that we only need a separate subset sum block for each input target neuron and not for its $\rho$ copies, as we can reuse each block in the construction of each output. $\square$

The advantage the construction above is that we can skip the approximation of the identity function. The real challenge, however, lies in the derivation of the required multiplicative factor $\rho$ for approximating a multi-layer target network.

## 2.3 Multi-layer Lottery Ticket Existence

Our main result is to derive realistically achievable lower bounds on the width of a source network that has depth $L_s = L_t + 1$. To prove the existence of LTs that approximate general multi-layer target networks using the last theorems, we have to overcome two challenges. First, when we consider multiple network layers, we need to understand how much error we can afford to make in each target parameter approximation for general activation functions beyond RELUs to stay below the global error bound, as error can get amplified when signal is sent through multiple layers. Second, we need to identify the required layer-wise width scaling factor $\rho$ in Thm. 2.7.

**Lemma 2.8** (Error propagation). *Let two networks $f_1$ and $f_2$ of depth $L$ have the same architecture and activation functions with Lipschitz constant $T$. Define $M_l := \sup_{x \in \mathcal{D}} \|\boldsymbol{x}^{(l)}\|_1$. Then, for any $\epsilon > 0$ we have $\|f_1 - f_2\|_\infty \leq \epsilon$, if every parameter $\theta_1$ of $f_1$ and corresponding $\theta_2$ of $f_2$ in layer $l$ fulfils $|\theta_1 - \theta_2| \leq \epsilon_l$ for*

$$\epsilon_l := \frac{\epsilon}{n_l L}\Big[T^{L-l+1}(1+M_{l-1})\Big(1+\frac{\epsilon}{L}\Big)\prod_{k=l+1}^{L-1}\Big(\big\|W_1^{(k)}\big\|_\infty + \frac{\epsilon}{L}\Big)\Big]^{-1}.$$

For suitable target networks, this requirement essentially becomes $\epsilon_l = C\epsilon/(n_l L)$. This is an advantageous estimate in comparison with [1], which derives a smaller allowed error $\epsilon_l \propto \epsilon/(N_t \prod_{s=l}^{L} N_t^{(s)})$ that is anti-proportional to the total number of nonzero parameters $N_t$ and a product over nonzero parameters in the upper layers. Our estimate can be used in two-layers-for-one as well as one-layer-for-one LT constructions. The latter follows as our main result.

**Theorem 2.9** (LT existence ($L+1$ construction)). *Assume that $\epsilon, \delta \in (0,1)$, a target network $f_t(x) : \mathcal{D} \subset \mathbb{R}^{n_0} \to \mathbb{R}^{n_L}$ with architecture $\bar{n}_t$ of depth $L_t$, $N_t$ nonzero parameters, and a source network $f_s$ with architecture $\bar{n}_s$ of depth $L_s = L_t + 1$ are given. Let $\phi_t$ be the activation function of $f_t$ in the layers $l \geq 2$ of $f_s$ with Lipschitz constant $T$, $\phi_0$ be the activation function of the first layer of $f_0$ fulfilling Assumption 2.3, and $M := \max\{1, \max_{\mathbf{x} \in \mathcal{D}, l} \big\|\mathbf{x}_t^{(1)}\big\|_1\}$. Let the parameters of $f_s$ be initialized according to Ass. 2.1 for $l > 2$ and Thm. 2.5 or 2.6 for $l \leq 2$. Then, with probability at least $1 - \delta$, $f_s$ contains a subnetwork $f_\epsilon \subset f_s$ so that each output component $i$ is approximated as $\max_{\boldsymbol{x} \in \mathcal{D}} |f_{t,i}(\boldsymbol{x}) - f_{\epsilon',i}(\boldsymbol{x})| \leq \epsilon$ if*

$$n_{s,l+1} \geq Cn_{t,l} \log\left(\frac{1}{\min\{\epsilon_{l+1}, \delta/\rho\}}\right)$$

*for $l \geq 1$, where $\epsilon_{l+1}$ is defined by Lemma 2.8 and $\rho = CN_t^{1+\gamma} \log(1/\min\{\min_l \epsilon_l, \delta\})$ for any $\gamma > 0$. Furthermore, we require $n_{s,1} \geq Cn_{t,1} \log\left(\frac{1}{\min\{\epsilon_{l+1}, \delta/\rho\}}\right)$.*

The full proof is presented in the appendix. While the layer-wise constructions are explained by the proofs of Thms. 2.6, 2.5, and 2.7, yet with updated $\epsilon_l$, the main challenge is to determine the increased number of subset sum problems that have to be solved to derive the scaling factor $\rho$. Note that $\rho$ only enters the logarithm and is negligible. By updating $C$, we in fact have the same asymptotic dependence $n_{s,l+1} \geq Cn_{t,l} \log(n_{t,l}L/\min\{\epsilon, \delta\})$ as in the two-layers-for-one construction.

# 3 Experiments

To demonstrate that our theoretical results make realistic claims, we present three sets of experiments that highlight different advantages of the $L + 1$-construction and the $2L$-construction. In all cases, we emulate our constructive existence proofs by pruning source networks to approximate a given target network. All experiments were conducted on a machine with Intel(R) Core(TM) i9-10850K CPU @ 3.60GHz processor and GPU NVIDIA GeForce RTX 3080 Ti.

Table 1: LT pruning results on MNIST. Averages and $0.95$ standard confidence intervals are reported for 5 independent source network initializations. Parameters are counted in packs of 1000.

| CONSTR. | TARGET | | $L + 1$ | | $2L$ | |
|---|---|---|---|---|---|---|
| | % ACC. | # PARAM. | % ACC. | # PARAM. | % ACC. | # PARAM. |
| RELU | 97.99 | 18.6 | $97.78 \pm 0.05$ | $1106.5 \pm 0.9$ | $97.96 \pm 0.02$ | $119.2 \pm 0.04$ |
| LRELU | 97.88 | 18.6 | $97.63 \pm 0.08$ | $1102.4 \pm 0.9$ | $97.84 \pm 0.06$ | $119.2 \pm 0.1$ |
| TANH | 98.2 | 18.6 | $98.07 \pm 0.07$ | $660.3 \pm 0.4$ | $98.14 \pm 0.03$ | $67.0 \pm 0.06$ |
| SIGMOID | 98.08 | 18.6 | $98.08 \pm 0.02$ | $669.7 \pm 0.4$ | $98.07 \pm 0.02$ | $67.4 \pm 0.05$ |

**LeNet on MNIST**    As our proofs suggest, pruning involves solving multiple subset sum approximation problems. Each is an NP-hard problem in general, as the size of the power set and thus the number of potential solutions scales exponentially in the base set size $m$, i.e., as $2^m$. However, as a set size of $m = 15$ and even smaller is sufficient for our purpose, we could solve each problem optimally by exhaustively evaluating all $2^m$ solutions. To reduce the size of the associated LTs, we instead identify the smallest subset consisting of up to 10 variables out of $m = 20$ to achieve an approximation error that does not exceed $0.01$.

What should be our target network? As the influential work [13], we use Iterative Magnitude Pruning (IMP) on LeNet networks with architecture $[784, 300, 100, 10]$ to find LTs that achieve a good performance on the MNIST classification task [7]. Using the Pytorch implementation of the Gihub repository open_lth[1] with MIT license, we arrive at a target network for each of four considered activation functions after 12 pruning steps: RELU, LRELU, SIGMOID, and TANH. Their performance and number of nonzero parameters are reported in Table 1 in the target column alongside our results for the $(L + 1)$-construction and our $2L$ construction, which achieve a similar performance. Note that, while the $L + 1$ construction relies in this case on a higher number of parameters, it uses less neurons and a smaller depth, which are the relevant criteria for fast computations and network training.

**ResNet18 on Tiny-ImageNet**    We have obtained more large-scale target networks by fine-tuning a ResNet18 model that was originally trained on ImageNet data and available for download on Pytorch for transfer learning. We replaced the last fully-connected classification layer by a fully-connected network with widths $[512, 512, 200]$ and activation function of interest in the first layer and trained the full ResNet model on the tiny-ImageNet training data. Similarly to experiments of [1], we estimated the pruning error of LTs for the fully-connected classification network based on a statistic of solving $10^5$ different subset sum approximation problems. The subset sum base set size in the one-layer-for-one construction is $m = 10$, while we used $m = 15$ in the two-layer-for-one construction. Note that this difference is possible because the subset sum base set consisting of uniformly distributed random variables in the one-layer-for-one construction can be smaller than variables that are products of uniform random variables as in the two-layers-for-one construction.

Both constructions approximate target parameters up to maximum error $\epsilon_l = 0.001$. The reported performance is evaluated on the tiny-ImageNet test data for a model that concatenates the residual target layers and a fully-connected LTs (see Table 2). In this case, because of the different base set sizes, we sometimes find the $L + 1$-construction to be more parameter efficient.

**Representational Benefits of the $L + 1$-construction**    In the previous experiments, we have constructed one target network and used different source networks to demonstrate the validity of our proofs and constructions. The more realistic set-up is, however, that the source network is given

---

[1]https://github.com/facebookresearch/open_lth

Table 2: LT pruning results on tiny-ImageNet. Averages and $0.95$ standard confidence intervals are reported for $3$ independent source network initializations. Parameters are counted in packs of $10^5$.

| CONSTR. | TARGET | | $L+1$ | | $2L$ | |
|---|---|---|---|---|---|---|
| | % ACC. | # PARAM. | % ACC. | # PARAM. | % ACC. | # PARAM. |
| RELU | 73.08 | 2.06 | $73.09 \pm 0.04$ | $19.14 \pm 0.005$ | $73.06 \pm 0.05$ | $21.8 \pm 0.004$ |
| LRELU | 73.0 | 2.06 | $72.96 \pm 0.04$ | $19.14 \pm 0.01$ | $72.92 \pm 0.03$ | $21.8 \pm 0.006$ |
| TANH | 73.73 | 2.06 | $73.75 \pm 0.04$ | $11.36 \pm 0.01$ | $73.72 \pm 0.08$ | $10.98 \pm 0.01$ |
| SIGMOID | 72.69 | 2.1 | $72.69 \pm 0.01$ | $19.14 \pm 0.002$ | $72.67 \pm 0.06$ | $21.86 \pm 0.01$ |

and thus the depth of the LT is predetermined as $L_s - 1$ or $L_s/2$, respectively. As a consequence, the $2L$- and the $L+1$-construction would approximate different target network representations. If the target of depth $L_t = L_s - 1$ is much sparser than the target network of depth $L_t = L_s/2$, our $L+1$-construction might be more effective than our $2L$-construction. The general challenge with this argument is that we cannot exclude the case that there might exist a much sparser target network of depth $L_t = L_s/2$ than we thought or could identify.

Keeping this caveat in mind, we have still constructed two types of target networks for a problem for which [11] has derived a relatively sparse solution. The circle target of depth $L_t = 25$ (for the $L+1$-construction) consists of 190 nonzero parameters and has a maximum width of 16, while the $L_t = 13$ target network (for the $2L$-construction) consists of 2133 parameters and has a maximum width of 1024. Both target networks are equipped with ReLU activation functions and achieve a similar test accuracy: $99.86\%$ is achieved by the $L_t = 13 = L_s/2$ target and $99.76\%$ by the $L_t = 25 = L_s - 1$ target.

In the following we report the properties of the lottery ticket that we identify by pruning the source network with a subset sum base set size of $m = 15$ in the two layer construction and base set size $m = 10$ in the one layer construction as averages over 3 independent runs with 0.95 standard significance intervals.

$2L$**-construction:** Acc: $99.5 \pm 0.6$, number of parameters: $98393 \pm 288$, maximum width: 15375.

$L+1$**-construction:** Acc: $99.8 \pm 0.01$, number of parameters: $18891 \pm 133$, maximum width: 170.

In this example, the $L+1$-construction results in a sparser lottery ticket consisting of fewer parameters and considerably smaller maximum width. The reason is that we could leverage almost the full depth of source network to obtain a much sparser target network representation for the $L+1$-construction.

## 4 Discussion

We have shown that randomly initialized fully-connected feed forward neural networks contain lottery tickets with high probability for a wide class of activation functions by deriving two types of constructions: (a) The $(2L_t)$-construction assumes that the larger random source network has at least double the depth of the target network and is wider only by a logarithmic factor $O(n_t \log(n_t L_t / \min\{\delta, \epsilon\})$ in the approximation error and the LT existence probability. (b) The $(L_t + 1)$-construction allows for potentially sparser target network representations, as these can utilize almost the full available depth $L_s = L_t + 1$ of the source network. Remarkable about this result is that asymptotically, we can maintain the logarithmic overparametrization. While this suggests a slightly less advantageous scaling in $\delta$ for the $(L_t + 1)$-construction, the constant is smaller for the $(L_t + 1)$-construction and most of the time we can choose a similar or smaller source width than in the $(2L_t)$-construction. We have also demonstrated in experiments that the LT for the $(L_t = L_s - 1)$-construction will often consist of fewer parameters than the LT corresponding to the $(L_s/2)$-construction, but this does not always have to be the case.

We have mainly discussed the scenario, in which the source network has exactly depth $L_s = L_t + 1$, but not all target network representations become sparser with increasing depth. What if the sparsest target representation has depth $L_t + 1 < L_s$? With the help of Lemma 2.4, we could simply add layers that approximate the identity and thus construct a target with depth $L_t = L_s - 1$. In future, it could be interesting to leverage excessive depth to distribute subset sum blocks on multiple layers instead, as it has been proposed for RELUS [3].

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
