# A   Subset Sum Approximation

In the discussed LT constructions, we generally have multiple random neurons and parameters available to approximate a target parameter $z$ by $\hat{z}$ up to error $\epsilon$ so that $|z - \hat{z}| \leq \epsilon$. Let us denote these random parameters in the source network as $X_i$. If these contain a uniform distribution, as defined below, we can utilize a subset of them for approximating $z$.

**Definition A.1.** A random variable $X$ contains a uniform distribution if there exist constants $\alpha \in (0, 1]$, $c, h > 0$ and a distribution $G_1$ so that $X$ is distributed as $X \sim \alpha U[c - h, c + h] + (1 - \alpha)G_1$.

[3] extended results by [30] to solve subset sum approximation problems if the random variables are not necessarily identically distributed. In addition, they also cover the case $|z| > 1$. The general statement follows below.

**Corollary A.2** (Subset sum approximation [30, 3]). *Let $X_1, ..., X_m$ be independent bounded random variables with $|X_k| \leq B$. Assume that each $X_k \sim X$ contains a uniform distribution with potentially different $\alpha_k > 0$ (see Definition A.1) and $c = 0$. Let $\epsilon, \delta \in (0, 1)$ and $t \in \mathbb{N}$ with $t \geq 1$ be given. Then for any $z \in [-t, t]$ there exists a subset $S \subset [m]$ so that with probability at least $1 - \delta$ we have $|z - \sum_{k \in S} X_k| \leq \epsilon$ if*

$$m \geq C \frac{\max\left\{1, \frac{t}{h}\right\}}{\min_k\{\alpha_k\}} \log\left(\frac{B}{\min\left(\frac{\delta}{\max\{1, t/h\}}, \frac{\epsilon}{\max\{t, h\}}\right)}\right).$$

In the two-layers-for-one construction, each $X_k$ is given by a product $X_k = w_{0,ik}^{(l+1)} w_{0,kj}^{(l)}$. Products of uniform distributions or normal distributions generally contain a uniform distribution [37]. However, $\alpha_i < 1$ is smaller for such products. In consequence, we can utilize a higher fraction of available parameters in case of a one-layer-for-one construction in comparison with a two-layers-for-one construction. Yet, this insight only affects the universal constant and is thus of minor influence.

# B   Proofs

## B.1   Proof of Lemma 2.4

*Statement* (Representation of the identity). For any $\epsilon' > 0$, for a function $\phi(x)$ that fulfills Assumption 2.3 with $a = a(\epsilon') > 0$, and for every $x \in [-a, a]$ we have

$$\left|x - \frac{1}{m_+ + m_-}\left(\phi(x) - \phi(-x)\right)\right| \leq \frac{2\epsilon'}{m_+ + m_-}. \tag{2}$$

*Proof.* According to Assumption 2.3, for any $x \in [-a(\epsilon'), a(\epsilon')]$ we have $|\phi(x) - (\mu_\pm(x)x + d)| \leq \epsilon'$, where $\mu_\pm(x) = m_+ x$ if $x \geq 0$ and $\mu_\pm(x) = m_- x$ otherwise. It follows that

$$|\phi(x) - \phi(-x) - (\mu_\pm(x)x + d - \mu_\pm(-x)x - d)| \leq |\phi(x) - \mu_\pm(x)x + d)| \\ + |\phi(-x) - \mu_\pm(-x)x + d| \leq 2\epsilon'. \tag{3}$$

Note that $\mu_\pm(x)x + d - \mu_\pm(-x)x - d = \mu_\pm(x) - \mu_\pm(-x) = (m_+ + m_-)x$. Thus, dividing Eq. (3) by $(m_+ + m_-)$ proves the statement. □

## B.2   Proof of Theorem 2.5

*Statement* (LT Existence (Two-for-One)). Assume that $\epsilon', \delta' \in (0, 1)$, a target network layer $f_t(x) : \mathcal{D} \subset \mathbb{R}^{n_0} \to \mathbb{R}^{n_1}$ with $f_{t,i}(x) = \phi_t\left(\sum_{j=1}^{n_0} w_{t,ij}x_j + b_{t,i}\right)$, and two source network layers $f_s$ are given with architecture $[n_0, n_{s,1}, n_1]$ and activation functions $\phi_0$ in the first and $\phi_t$ in the second layer. Let $\phi_0$ fulfill Assumption 2.3 with $a > 0$ and $d = 0$, $\phi_t$ have modulus of continuity $\omega_t$, and $M := \max\{1, \max_{\mathbf{x} \in \mathcal{D}, i} |x_i|\}$. Let the parameters of $f_s$ be conveniently initialized as $w_{ij}^{(1)}, b_i^{(1)} \sim U[-\sigma, \sigma]$ and $w_{ij}^{(2)} \sim U[-1/(|m_+ + m_-|\sigma), 1/(|m_+ + m_-|\sigma)]$, $b_i^{(2)} = 0$, where $\sigma = \min\{1, a(\epsilon'')/M\}$ with $\epsilon'' = g^{-1}\left(\omega_t^{-1}(\epsilon')/\left(Cn_0 \frac{M}{|m_+ + m_-|} \log\left(\frac{n_0}{\min\{\delta'/n_{t,1}, \omega_t^{-1}(\epsilon')/M\}}\right)\right)\right)$.

Then, with probability at least $1 - \delta'$, $f_s$ contains a subnetwork $f_{\epsilon'} \subset f_s$ so that each output component $i$ is approximated as $\max_{\boldsymbol{x} \in \mathcal{D}} |f_{t,i}(\boldsymbol{x}) - f_{\epsilon',i}(\boldsymbol{x})| \leq \epsilon'$ if

$$n_{s,1} \geq C n_0 \log \left( \frac{n_0}{\min\{\omega_t^{-1}(\epsilon')/M, \delta'/n_{t,1}\}} \right).$$

*Proof.* The construction of a LT consists of three main steps. First, we prune the hidden neurons of $f_s$ to univariate form. Second, we identify neurons in the hidden layer for which we can approximate $\phi_0$ for small inputs according to Assumption 2.3. Third, if $n_{s,1}$ is large enough, we can select subsets $I_j$ and $I_b$ of the hidden neurons with small inputs so that we can use Thm. A.2 on subset sum approximation to approximate the parameters of the target network. The resulting subnetwork of $f_s$ is of the following form $f_{\epsilon',i}(x) = \phi_t \left( \sum_{k \in I} w_{ik}^{(2)} \phi_0 \left( w_{kj}^{(1)} x_j \right) + \sum_{k \in I_b} w_{ik}^{(2)} \phi_0 \left( b_k^{(1)} \right) \right)$, where $I = \cup_j I_j$ and $\sum_{k \in I_j} w_{ik}^{(2)} w_{kj}^{(1)}$ can be used to approximate $w_{t,ij}$. $\lambda f_{\epsilon,i}$ qualifies as LT if

$$|f_{t,i}(x) - \lambda f_{\epsilon',i}(x)| = |\phi_t(x_{t,i}^{(1)}) - \phi_t(x_{\epsilon',i}^{(1)})| \leq \omega_t \left( |x_{t,i}^{(1)} - x_{\epsilon',i}^{(1)}| \right) \leq \epsilon', \tag{4}$$

where the last inequality holds if we can show that $|x_{t,i}^{(1)} - x_{\epsilon',i}^{(1)}| \leq \omega_t^{-1}(\epsilon')$. Thus, let us bound

$$|x_{t,i}^{(1)} - x_{\epsilon',i}^{(1)}| = \left| \sum_{j=1}^{n_0} \left[ w_{t,ij} x_j - \sum_{k \in I_j} w_{ik}^{(2)} \phi_0 \left( w_{kj}^{(1)} x_j \right) \right] + \left[ b_{t,i} - \sum_{k \in I_b} w_{ik}^{(2)} \phi_0 \left( b_k^{(1)} \right) \right] \right|$$
$$\leq n_0 \max_{x \in \mathcal{D}, j} \left| w_{t,ij} x_j - \sum_{k \in I_j} w_{ik}^{(2)} \phi_0 \left( w_{kj}^{(1)} x_j \right) \right| + \left| b_{t,i} - \sum_{k \in I_b} w_{ik}^{(2)} \phi_0 \left( b_k^{(1)} \right) \right|. \tag{5}$$

Thus, we have to ensure that each summand

$$\left| w_{t,ij} x_j - \sum_{k \in I_j} w_{ik}^{(2)} \phi_0 \left( w_{kj}^{(1)} x_j \right) \right| \leq \frac{\omega_t^{-1}(\epsilon')}{(n_0 + 1)}. \tag{6}$$

This also applies to the approximation of $b_{t,i}$, respectively. To achieve this, we would like to approximate $\phi_0 \left( w_{kj}^{(1)} x_j \right) \approx \mu_{\pm}(w_{kj}^{(1)} x_j) w_{kj}^{(1)} x_j$, where $\mu_{\pm}(x) = m_+ x$ if $x \geq 0$ and $\mu_{\pm}(x) = m_- x$ otherwise. This is valid, as $|w_{kj}^{(1)} x_j| \leq M a(\epsilon'')/M = a(\epsilon'')$ by construction, as long as we pick $\epsilon''$ small enough. We thus obtain

$$\left| w_{t,ij} x_j - \sum_{k \in I_j} w_{ik}^{(2)} \phi_0 \left( w_{kj}^{(1)} x_j \right) \right|$$
$$\leq \underbrace{\left| w_{t,ij} x_j - \sum_{k \in I_j} \mu_{\pm}(w_{kj}^{(1)} x_j) w_{ik}^{(2)} w_{kj}^{(1)} x_j \right|}_{\leq \frac{\omega_t^{-1}(\epsilon')}{2(n_0+1)} \text{ by subset sum approx.}} + \underbrace{\left| \sum_{k \in I_j} w_{ik}^{(2)} \left( \phi_0 \left( w_{kj}^{(1)} x_j \right) - \mu_{\pm}(w_{kj}^{(1)} x_j) w_{kj}^{(1)} x_j \right) \right|}_{\leq \frac{\omega_t^{-1}(\epsilon')}{2(n_0+1)} \text{ by activation funct. approx.}} \tag{7}$$

Let us first focus on the subset sum approximation. It helps to split the index set $I_j$ into indices $I_j^+$ for which $w_{kj}^{(1)} > 0$ and $I_j^-$ all the ones for which $w_{kj}^{(1)} < 0$. Furthermore, note that if $w_{kj}^{(1)} > 0$ it holds that $\mu_{\pm}(w_{kj}^{(1)} x_j)/(m_+ + m_-) = |\mu_{\pm}(x_j)/(m_+ + m_-)| = 1 - |\mu_{\pm}(-x_j)/(m_+ + m_-)|$. In

consequence, we have

$$\left| w_{t,ij} x_j - \sum_{k \in I_j} \mu_\pm(w_{kj}^{(1)} x_j) w_{ik}^{(2)} w_{kj}^{(1)} x_j \right| \leq M \left| w_{t,ij} - \sum_{k \in I_j} \mu_\pm(w_{kj}^{(1)} x_j) w_{ik}^{(2)} w_{kj}^{(1)} \right|$$

$$\leq M \frac{|\mu_\pm(x_j)|}{|m_+ + m_-|} \underbrace{\left| w_{t,ij} - |m_+ + m_-| \sum_{k \in I_j^+} w_{ik}^{(2)} w_{kj}^{(1)} \right|}_{\leq \frac{\omega_t^{-1}(\epsilon')}{2M(n_0+1)}}$$

$$+ M \left( 1 - \frac{|\mu_\pm(x_j)|}{|m_+ + m_-|} \right) \underbrace{\left| w_{t,ij} - |m_+ + m_-| \sum_{k \in I_j^-} w_{ik}^{(2)} w_{kj}^{(1)} \right|}_{\leq \frac{\omega_t^{-1}(\epsilon')}{2M(n_0+1)}}$$

Thm. A.2 can guaranty a small enough error with probability $1 - \delta'$. As we have to solve two subset sum approximation problems per parameter and thus $2(n_0 + 1)n_{t,1}$ in total, we need to solve each of them successfully with probability at least $1 - \delta'/(2(n_0 + 1)n_{t,1})$, which can be seen with the help of a union bound. The random variables that are used in the approximation are given by $X_k = |m_+ + m_- | w_{ik}^{(2)} w_{kj}^{(1)}$. The initial scaling of the random variables is exactly chosen so that $X_k$ is given by the product of two uniform random variables $X_k \sim U[-1,1]U[0,1]$ or $X_k \sim U[-1,1]U[-1,0]$, which contain a normal distribution as shown by [36]. Therefore, Thm. A.2 states that if

$$n \geq C \log \left( \frac{n_0}{\min \left\{ \delta'/n_{t,1}, \omega_t^{-1}(\epsilon')/M \right\}} \right) \tag{8}$$

random variables are available for each subset sum approximation and thus $n \geq Cn_0 \log \left( n_0 / \min\{\delta, \omega_t^{-1}(\epsilon')/M\} \right)$ in total, we can achieve the desired subset sum approximation error.

It is left to show how we can obtain the necessary activation function approximation in Eq. (7). The relevant term vanishes for $a(\epsilon'') = \infty$ and the claim follows directly. In the following, we discuss therefore only the case that $a(\epsilon'')$ is finite.

$$\left| \sum_{k \in I_j} w_{ik}^{(2)} \left( \phi_0 \left( w_{kj}^{(1)} x_j \right) - \mu_\pm(w_{kj}^{(1)} x_j) w_{kj}^{(1)} x_j \right) \right| \leq \sum_{k \in I_j} \left| w_{ik}^{(2)} \right| \left| \phi_0 \left( w_{kj}^{(1)} x_j \right) - \mu_\pm(w_{kj}^{(1)} x_j) w_{kj}^{(1)} x_j \right|$$

$$\leq \sum_{k \in I_j} \left| w_{ik}^{(2)} \right| \epsilon'' \leq \frac{|I_j|}{|m_+ + m_-|\sigma} \epsilon'' \leq C \log \left( \frac{n_0}{\min \left\{ \delta, \omega_t^{-1}(\epsilon')/M \right\}} \right) \frac{M}{|m_+ + m_-|} \frac{\epsilon''}{a(\epsilon'')},$$

where we have used that $\left| w_{ik}^{(2)} \right| \leq \frac{1}{|m_+ + m_-|\sigma}$ with $\sigma = \frac{a(\epsilon'')}{M}$ and the fact that the approximation of the activation function is valid with $\epsilon''$ by construction. Note that the required number of subset elements $|I_j|$ in the subset sum approximation is usually much smaller than the used upper bound of the whole set size as given by Eq. (8). To arrive at the end of the proof, we only have to choose $\epsilon''$ small enough so that the whole term is bounded by $\frac{\omega_t^{-1}(\epsilon')}{2(n_0+1)}$. Interestingly, this choice only affects the scaling in the initialization of the random variables but not directly our width requirement. We can always find an appropriate $\epsilon''$ and accordingly also scaling factor $\sigma = \frac{a(\epsilon'')}{M}$, as the function $g(\epsilon'') = \frac{\epsilon''}{a(\epsilon'')}$ is invertible on a suitable interval $]0, \epsilon']$. We can therefore define

$$\epsilon'' = g^{-1} \left( \frac{\omega_t^{-1}(\epsilon')}{Cn_0 \log \left( \frac{n_0}{\min\{\delta'/n_{t,1}, \omega_t^{-1}(\epsilon')/M\}} \right) \frac{M}{|m_+ + m_-|}} \right). \tag{9}$$

$\square$

We also need that $\lim_{\epsilon'' \to 0} g(\epsilon'') = 0$ to make sure that we can always find a suitable $\epsilon''$ for any choice of $\epsilon'$. Furthermore, note that, if $m_+ = m_- = m$, we do not need to distinguish $w_{kj}^{(1)} > 0$ and $w_{kj}^{(1)} < 0$ to do separate approximations. In this case, we only need to solve $n_0 + 1$ subset sum approximation problems.

To give an example for $g$, let us recall that for tanh (and sigmoids) we have $a(\epsilon'') = C\epsilon''^{1/3}$. In consequence, $g(\epsilon'') = \frac{\epsilon''}{a(\epsilon'')} = C\epsilon''^{2/3}$ and thus $g^{-1}(y) = Cy^{3/2}$ so that $\epsilon''$ is of order $\epsilon'' = C\left(\epsilon'/\log(1/\epsilon')\right)^{2/3}$ in this case.

## B.3 Proof of Thm. 2.6

*Statement* (LT Existence (Two-for-One) with Non-Zero Intercept). Thm. 2.5 applies also to activation functions $\phi_0$ that fulfill Assumption 2.3 with $d \neq 0$ if the parameters are initialized according to Assumption 2.2 with $M_0^{(l)}$ distributed as the weights in Thm. 2.5.

*Proof.* We can closely follow the steps of the previous proof. The major difference is that we approximate $\phi_0(x) \approx \mu_\pm(x)x + d$. This turns the activation function approximation in Eq. 7 into

$$
\left| w_{t,ij}x_j - \sum_{k \in I_j} w_{ik}^{(2)}\phi_0\left(w_{kj}^{(1)}x_j\right) \right| \leq \underbrace{\left| w_{t,ij}x_j - \sum_{k \in I_j} \mu_\pm(w_{kj}^{(1)}x_j)w_{ik}^{(2)}w_{kj}^{(1)}x_j \right|}_{\leq \frac{\omega_t^{-1}(\epsilon')}{2(n_0+1)} \text{ by subset sum approx.}} \tag{10}
$$

$$
+ \underbrace{\left| \sum_{k \in I_j} w_{ik}^{(2)}\left(\phi_0\left(w_{kj}^{(1)}x_j\right) - \mu_\pm(w_{kj}^{(1)}x_j)w_{kj}^{(1)}x_j - d\right) \right|}_{\leq \frac{\omega_t^{-1}(\epsilon')}{2(n_0+1)} \text{ by activation funct. approx.}} + \left| d \sum_{k \in I_j} w_{ik}^{(2)} \right|.
$$

In principle, we could have modified the bias subset sum approximation by approximating $b_{t,i} + d\sum_j \sum_{k \in I_j} w_{ik}^{(2)}$ instead of $b_{t,i}$. Yet, $\sum_j \sum_{k \in I_j} w_{ik}^{(2)}$ could be a large number, with which we would need to multiply our width requirement, if each $w_{ik}^{(2)}$ is initialized as in Thm. 2.5. In contrast, with the looks-linear initialization we can choose $w_{ik'}^{(2)} = -w_{ik}^{(2)}$ so that $\sum_{k \in I_j} w_{ik}^{(2)} = \sum_{k \in I_j^+} w_{ik}^{(2)} + \sum_{k' \in I_j^-} w_{ik'}^{(2)} = \sum_{k \in I_j^+} w_{ik}^{(2)} - \sum_{k \in I_j^+} w_{ik}^{(2)} = 0$. The extra term vanishes and we only need to solve half of the subset sum approximation problems than in the previous theorem, i.e. only $(n_0 + 1)$, with probability $\delta'/(n_0 + 1)$. Thus, we could also bound

$$
n \geq C \log\left( \frac{n_0}{\min\left\{\delta, \omega_t^{-1}(\epsilon')/(2M)\right\}} \right) \tag{11}
$$

with a smaller $C$ than in Thm. 2.5 but we do not derive the precise constant anyways. $\square$

Note that even in the case $m_+ = m_- = m$, if $d \neq 0$, we distinguish the cases $w_{kj}^{(1)} > 0$ and $w_{kj}^{(1)} < 0$ to do separate approximations, as this leads to a vanishing $\sum_{k \in I_j} w_{ik}^{(2)} = 0$.

## B.4 Proof of Lemma 2.8

*Statement* (Error propagation). Let two networks $f_1$ and $f_2$ of depth $L$ have the same architecture and activation functions with Lipschitz constant $T$. Define $M_l := \sup_{x \in \mathcal{D}} \left\| x_1^{(l)} \right\|_1$. Then, for any $\epsilon > 0$ we have $\|f_1 - f_2\|_\infty \leq \epsilon$, if every parameter $\theta_1$ of $f_1$ and corresponding $\theta_2$ of $f_2$ in layer $l$ fulfils $|\theta_1 - \theta_2| \leq \epsilon_l$ for

$$
\epsilon_l := \frac{\epsilon}{n_l L}\left[ T^{L-l+1}\left(1 + M_{l-1}\right)\left(1 + \frac{\epsilon}{L}\right) \prod_{k=l+1}^{L-1}\left(\left\|W_1^{(k)}\right\|_\infty + \frac{\epsilon}{L}\right) \right]^{-1}.
$$

*Proof.* Using the Lipschitz continuity of the activation function, we obtain for each component of the difference between the two networks

$$|f_{1,i}(x) - f_{2,i}(x)| = \left| \phi\left(\sum_j w^{(L)}_{1,ij} x^{(L-1)}_{1,j} + b^{(L)}_{1,i}\right) - \phi\left(\sum_j w^{(L)}_{2,ij} x(L-1)_{2,j} + b^{(L)}_{2,i}\right) \right|$$

$$\leq T \left| \sum_j (w^{(L)}_{1,ij} x^{(L-1)}_{1,j} - w^{(L)}_{2,ij} x(L-1)_{2,j}) + b^{(L)}_{1,i} - b^{(L)}_{2,i} \right| \leq T \sum_j |w^{(L)}_{1,ij} - w^{(L)}_{2,ij}||x^{(L-1)}_{1,j}|$$

$$+ T \sum_j |w^{(L)}_{2,ij}||x^{(L-1)}_{2,j} - x^{(L-1)}_{1,j}| + T|b^{(L)}_{1,i} - b^{(L)}_{2,i}|$$

$$\leq T \left[ \epsilon_L \left\|x^{(L-1)}_1\right\|_1 + (1+\epsilon_L) \left\|x^{(L-1)}_2 - x^{(L-1)}_1\right\|_1 + \epsilon_L \right]$$

$$\leq T \left[ \epsilon_L (1 + M_{l-1}) + (1+\epsilon_L)T \left\| W^{(L-1)}_2 x^{(L-2)}_2 + b^{(L-1)}_2 - W^{(L-1)}_1 x^{(L-2)}_1 - b^{(L-1)}_1 \right\| \right]$$

$$\leq \epsilon_L (1 + M_{l-1}) + (1+\epsilon_L)T^2 \left[ \left\| W^{(L-1)}_2 \left( x^{(L-2)}_2 - x^{(L-2)}_1 \right)\right\|_1 + \left\| \left( W^{(L-1)}_2 - W^{(L-1)}_1 \right) x^{(L-2)}_1 \right\|_1 \right.$$

$$+ \left\| b^{(L-1)}_2 - b^{(L-1)}_1 \right\|_1 \right] \leq \epsilon_L (1 + M_{l-1}) + (1+\epsilon_L)T^2 \left[ \left( \left\|W^{(L-1)}_1\right\|_\infty + n_{L-1}\epsilon_{L-1} \right) \right.$$

$$\times \left\| x^{(L-2)}_2 - x^{(L-2)}_1 \right\|_1 + n_{L-1}\epsilon_{L-1} \left\| x^{(L-2)}_1 \right\|_1 + n_{L-1}\epsilon_{L-1} \right]$$

$$\leq \epsilon_L (1 + M_{l-1}) + (1+\epsilon_L)T^2 n_{L-1}\epsilon_{L-1}(1 + M_{L-2}) + (1+\epsilon_L)T^2 \left( \left\|W^{(L-1)}_1\right\|_\infty + n_{L-1}\epsilon_{L-1} \right)$$

$$\times \left\| x^{(L-2)}_2 - x^{(L-2)}_1 \right\|_1 , \tag{12}$$

where we have assumed that each $|w^{(L)}_{1,ij}| \leq 1$. The above lemma further assumes that each parameter in layer $L$ of network 2 is maximally $|w^{(L)}_{2,ij}| \leq |w^{(L)}_{1,ij}| + \epsilon_L \leq 1 + \epsilon_L$. In addition, it claims that $\left\|x^{(L-1)}_1\right\|_1 \leq M_{L-1}$. Repeating the above arguments iteratively for $\left\|x^{(l)}_2 - x^{(l)}_1\right\|_1$, we arrive at the following bound

$$|f_{1,i}(x) - f_{2,i}(x)| \leq T\epsilon_L(1 + M_{L-1})$$
$$+ \sum_{l=1}^{L-1} T^{L-l+1}(M_{l-1} + 1)n_l\epsilon_l(1+\epsilon_L) \prod_{k=l+1}^{L-1} \left( \left\|W^{(k)}_1\right\|_\infty + n_k\epsilon_k \right). \tag{13}$$

We have to ensure that this expression is smaller or equal to $\epsilon$. This can by achieved by assigning to each term that is related to a layer $l$ the maximum error $\epsilon/L$. It follows that also $\epsilon_l \leq \epsilon/(Ln_l)$ so that

$$T^{L-l+1}(M_{l-1} + 1)n_l\epsilon_l(1+\epsilon_L) \prod_{k=l+1}^{L_1} \left( \left\|W^{(k)}_1\right\|_\infty + n_k\epsilon_k \right) \tag{14}$$

$$\leq T^{L-l+1}(M_{l-1} + 1)n_l\epsilon_l \left(1 + \frac{\epsilon}{L}\right) \prod_{k=l+1}^{L_1} \left( \left\|W^{(k)}_1\right\|_\infty + \frac{\epsilon}{L} \right) \leq \frac{\epsilon}{L} \tag{15}$$

Solving the last inequality for $\epsilon_l$ proves our claim. $\qquad\square$

## B.5 Proof of Theorem 2.9

*Statement* (LT existence ($L + 1$ construction)). Assume that $\epsilon, \delta \in (0, 1)$, a target network $f_t(x) : \mathcal{D} \subset \mathbb{R}^{n_0} \to \mathbb{R}^{n_L}$ with architecture $\bar{n}_t$ of depth $L$, $N_t$ non-zero parameters, and a source network $f_s$ with architecture $\bar{n}_0$ of depth $L + 1$ are given. Let $\phi_t$ be the activation function of $f_t$ and the layers $l \geq 2$ of $f_s$ with Lipschitz constant $T$, $\phi_0$ be the activation function of the first layer of $f_s$ fulfilling Assumption 2.3, and $M := \max\{1, \max_{\mathbf{x}\in\mathcal{D},l} \left\|\mathbf{x}^{(1)}_{\mathbf{t}}\right\|\}$. Let the parameters of $f_s$ be conveniently initialized according to Assumption 2.1 for $l \geq 2$ and Thm. 2.5 or 2.6 for $l \leq 1$. Then,

with probability at least $1 - \delta$, $f_s$ contains a subnetwork $f_\epsilon \subset f_s$ so that each output component $i$ is approximated as $\max_{\boldsymbol{x} \in \mathcal{D}} |f_{t,i}(\boldsymbol{x}) - f_{\epsilon',i}(\boldsymbol{x})| \leq \epsilon$ if for $l \geq 1$

$$n_{s,l+1} \geq Cn_{t,l} \log\left(\frac{1}{\min\{\epsilon_{l+1}, \delta/\rho\}}\right),$$

where $\epsilon_{l+1}$ is defined by Lemma 2.8 and $\rho = CN_t^{1+\gamma} \log(1/\min\{\min_l \epsilon_l, \delta\})$ for any $\gamma > 0$. Furthermore, we require $n_{s,1} \geq Cn_{t,1} \log\left(\frac{1}{\min\{\epsilon_{l+1}, \delta/\rho\}}\right)$.

*Proof.* The proofs of Thms. 2.6, 2.5, and 2.7 have already explained the main parts of the construction. The missing piece is the choice of appropriate modification of $\delta$ by $\rho \geq \rho' = \sum_{l=1}^{L} \rho'_l$, where $\rho'$ counts the increased number of required subset sum approximation problems to approximate the $L$ target layers with our lottery ticket and $\rho_l$ counts the same number just for Layer $l$.

For each non-zero parameter, we will need two solve at least one subset sum approximation problem or sometimes two in case of the first target layer. We denote the number of non-zero parameters in Layer $l$ as $N_l$. Thus, if our target network is fully-connected and all parameters are non-zero, we have $N_l = n_{t,l}(n_{t,l-1} + 1)$ and in total $N_t = \sum_{l=1}^{L} n_{t,l}(n_{t,l-1} + 1)$.

Let us start with counting the number $\rho'_L$ of required subset sum approximation problems in the last layer because it determines how many neurons we need in the previous layer. This in turn defines how many subset sum approximation problems we have to solve to construct this previous layer.

The last layer requires us to solve exactly $\rho'_L = N_L$ subset sum problems, which can be solved successfully with high probability if $n_{s,L-1} \geq Cn_{t,L-1} \log(1/\min\{\epsilon_L, \delta/\rho'\})$. We will only need to construct a subset of these neurons with the help of Layer $L - 2$, i.e., exactly the neurons that are used in the lottery ticket. If $n_{s,L-1}$ is large, this might require only $2 - 3$ neurons per parameter. For simplicity, however, we bound this number by the total number of available neurons. To reconstruct one set of neurons, we need approximate $N_{L-1}$ parameters. As we have to maximally construct $C \log(1/\min\{\epsilon_L, \delta/\rho'\})$ sets of these neurons, we can bound $\rho'_{L-1} \leq CN_{L-1} \log(1/\min\{\epsilon_L, \delta/\rho'\})$.

Note that we can solve all of these subset sum approximation problems with the help of $n_{t,L-2} \geq CN_{L-2} \log(1/\min\{\epsilon_{L-1}, \delta/\rho'\}$ neurons and this number does not increase by the fact that we have to construct not only $n_{t,L-1}$ neurons but a number of neurons that is increased by a logarithmic factor. The higher number of required neuron approximations only affects the number of required subset sum approximation problems and thus the needed success probability of each parameter approximation via $\rho$.

Repeating the same argument for every layer, we derive $\rho'_l \leq CN_l \log(1/\min\{\epsilon_{l+1}, \delta/\rho'\}$, which could also be shown formally by induction. In total we thus find $\rho' = \sum_{l=1}^{L} \rho'_l \leq CN_l \log(1/\min\{\min_l \epsilon_l, \delta/\rho'\}) \leq CN_t \log(1/\min\{\min_l \epsilon_l, \delta/\rho\})$. A $\rho$ that fulfills $\rho \geq CN_t \log(1/\min\{\min_l \epsilon_l, \delta/\rho\})$ would therefore be sufficient to prove our claim. $\rho = CN_t^{1+\gamma} \log(1/\min\{\epsilon, \delta\})$ for any $\gamma > 0$ works, as $CN_t^\gamma \geq \log(N_t)$. $\square$