# OpenReview forum: "Most Activation Functions Can Win the Lottery Without Excessive Depth"
_NeurIPS.cc/2022/Conference — NeurIPS 2022 Accept_

### Official Review · Reviewer_NsjM · 2022-06-18

**Rating:** 7
**Confidence:** 3
**Soundness:** 4 excellent
**Presentation:** 4 excellent
**Contribution:** 3 good

**Summary:**

This paper generalizes the lottery ticket hypothesis by (1) extending results beyond ReLU activation functions to a broad class and (2) improving the 2L depth to depth L+1.

**Questions:**

Is there a reason why practitioners should care about the depth 2L to L+1 improvement, or is the interest merely theoretical?  If it is of practical significance, please provide an experiment demonstrating so.

**Limitations:**

The main limitation is in the experiments, which are at best just a simple sanity check of the theory with a small network and dataset.  Since the paper focuses on flexibility (many activation functions) and efficiency (reducing depth from 2L to L+1) it would be much more impressive to see the technique applied in a more realistic setting with a larger network and dataset.

**Strengths And Weaknesses:**

- Originality: The originality of the paper is twofold: a different network construction allows for depth L+1 instead of 2L, and flexibility in the choice of activation function means this technique can potentially be applied to many more types of architectures.

- Quality: This is a good paper.  There is a clear discussion of the related work and how this paper relates to past papers, and this paper makes a concrete, incremental improvement over the past works.

- Clarity: The writing is clear and well-organized.  No complaints here.

- Significance: The main significance at this point are the theoretical contributions.  I believe there is a potential to make a stronger statement about the practical significance with additional experiments.

---

> ### Author Response · Authors · 2022-08-02
> **Answer to Reviewer NsjM**
>
> We thank Reviewer NsjM for the positive feedback and the helpful suggestion to strengthen our message. Below, we answer the question regarding the practicality of the L+1 construction and add more large scale experiments for ResNet18 on tiny-imagenet.
>
> 1. In general, it is quite uncommon to train very deep fully connected neural networks. Most performance benefits on standard benchmark datasets have been derived by concatenating convolutional, residual, or even transformer architectures. Often, only a single fully connected layer is used in the final classification layer. Yet, this is also the reason why practitioners should care about the L+1 construction, as it can explain the success of pruning a single fully connected layer (if it is combined with the previous layer, for instance, a convolutional layer). The 2L construction would require at least 2 fully-connected layers in the source network.
>
> 2. We would like to highlight that also our 2L construction is novel for activation functions that are different from ReLUs and can have advantages. However, we should have highlighted that LTs that are obtained by the $L+1$-construction usually have a smaller maximum width and consist of fewer total neurons in comparison with the $2L$-construction. To emphasize this point we have also reported these measures in the table that reports our additional experiments.
>
> 3. We have added the following table with additional experiments in our appendix. We have obtained the target networks by fine-tuning ResNet18 models that were originally trained on ImageNet data and available for download on Pytorch for transfer learning. We replaced the last fully-connected classification layer by a fully-connected network with widths $[512, 512, 200]$ and activation function of interest in the first layer and trained the full ResNet model on the tiny-ImageNet training data. Similarly to our experiments on MNIST, we have constructed lottery tickets that approximate the fully-connected classification network by pruning larger random source networks. The subset sum base set size in the one-layer-for-one construction is $10$, while we used $15$ in the two-layer-for-one construction. Both construct target parameters up to maximum approximation error $\epsilon_l = 0.001$. The reported performance is evaluated on the tiny-ImageNet test data for a model that concatenates the residual target layers and a fully-connected lottery ticket.
>
> LT pruning results on tiny-ImageNet. Averages and $0.95$ standard confidence intervals are reported for $3$ independent source network initializations. Parameters are counted in packs of $10^5$:
> |             LT constr.          |     |          Target     |                |   $L+1$      |                |        $2L$    |
> | ------------- |:-------------:| -----:| -----:| -----:| -----:| -----:|
> |   Act. funct.   |  \% Acc.  |  Param.  | \% Acc.  |  Param.  |  \% Acc.  |  Param.  |
>  ReLU   | 73.08 | 2.06 | 73.09 $\pm$ 0.04  | 19.14 $\pm$ 0.005 |  73.06 $\pm$ 0.05 | 21.8 $\pm$ 0.004
>  LReLU  | 73.0 | 2.06 | 72.96 $\pm$ 0.04 | 19.14 $\pm$ 0.01 | 72.92 $\pm$ 0.03 | 21.8 $\pm$ 0.006
> Tanh  | 73.73 | 2.06 | 73.75 $\pm$ 0.04 | 11.36 $\pm$ 0.01 | 73.72 $\pm$ 0.08 | 10.98 $\pm$ 0.01
> Sigmoid | 72.69 | 2.1 | 72.69 $\pm$ 0.01 | 19.14 $\pm$ 0.002 | 72.67 $\pm$ 0.06 | 21.86 $\pm$ 0.01
>
> Note that in these experiments the LTs of the $L+1$-construction and $2L$-construction consist of a similar number of total parameters. The reason is that we reduced the subset sum approximation base set size in the one-layer-for-one construction, as this is feasible.

---

> > ### Author Response · Authors · 2022-08-02
> > **Representational benefits by increased target depth**
> >
> > 4. How could we demonstrate the effectiveness of the L+1 construction?
> > Ideally, we would like to start from the same random source network of depth $L_s$ and approximate either a target network of depth $L_t = L_s/2$ or $L_t = L_s-1$ depending on the construction. If the target of depth $L_t = L_s-1$ is much sparser than the target network of depth $L_t = L_s/2$, our $L+1$-construction might be more effective than our $2L$-construction. The general challenge with this argument is that we cannot exclude the case that there might exist a much sparser target network of depth $L_t = L_s/2$ than we thought or could identify. Keeping this caveat in mind, we have still constructed two types of target networks for a problem for which [12] has derived a relatively sparse solution. The circle target of depth $L_t = 25$ (for the $L+1$-construction) consists of $190$ nonzero parameters and has a maximum width of $16$, while the $L_t = 13$ target network (for the $2L$-construction) consists of $2133$ parameters and has a maximum width of $1024$. Both target networks are equipped with ReLU activation functions and achieve a similar test accuracy: 99.86\% is achieved by the $L_t = 13=L_s/2$ target and 99.76\% by the  $L_t = 25 =L_s-1$ target.
> >
> > In the following we report the properties of the lottery ticket that we identify by pruning the source network with a subset sum base set size of 15 in the two layer construction and base set size 10 in the one layer construction as averages over 3 independent runs with 0.95 standard significance intervals.
> >
> > **$2L$-construction:**  Acc: $99.5 \pm 0.6$, Nbr. params: $98393 \pm 288$, Max width: 15375.
> >
> > **$L+1$-construction:** Acc: $99.8 \pm 0.01$, Nbr. params: $18891 \pm 133$, Max width: 170.
> >
> > In this example, the $L+1$-construction results in a sparser lottery ticket consisting of fewer parameters and considerably smaller maximum width. The reason is that we could leverage almost the full depth of source network to obtain a much sparser target network representation for the $L+1$-construction.

---

### Official Review · Reviewer_Puqv · 2022-07-11

**Rating:** 2
**Confidence:** 3
**Soundness:** 1 poor
**Presentation:** 2 fair
**Contribution:** 1 poor

**Summary:**

The authors attempt to extend previous work proving the existence of "Strong Lottery Tickets", strong LTs, i.e. subnetworks of a randomly-initialized neural network that show moderate generalization without training. The previous work had proven the existence of strong LTs of depth L within randomly initialized source neural networks of depth 2L, with the restriction of using ReLU activation functions. The authors claim to show a proof of such an existence within a source neural network of depth L+1, and assuming a broader class of activation functions. The authors attempt to validate their claim by finding weak LTs (i.e. LTs from a trained/pruned neural network) in source models of depth 2L and L+1, and comparing the generalization of the weak LTs.

**Questions:**

* 26-34: unclear, some sentences do not make sense at all -- needs a rewrite
* 38-40: how do these function approximation results relate to sparse nets?
* define strong and weak/strong Lottery Ticket Hypothesis, and weak/strong Lottery Tickets in the beginning (currently there are some definitions in the second paragraph of related work, but these need clarification also)
* it should be emphasized that the target network is trained (at least this is what I have inferred, but I am still unsure!)
Related Literature: all of this should be put in a relation to the topic of your work; the second paragraph explains the topic of your work and should rather be in the intro
* 67-69: “Their existence has been proven [...] by providing lower bounds on the width of the large, randomly initialized source network” -- bounding the width is not an existence proof. proven theoretically or shown empirically? if theoretically, what method was used?
* 80: "we usually care about cases where \epsilon < \delta" \epsilon is approximation error, and \delta is 1-probability, how/why would we compare these? It's apples v.s. oranges from what I can see.
* 87-88: Why is a bound on the weight magnitude not a loss of generality?
* 95-98: “we assume a convenient parameter initialization that we have to choose with respect to the activation function if we approximate a target layer with two source network layers”
    1) how is the initialization related to the activation function?
    2) why is a target layer approximated with two source-net layers? is this an assumption, or is this enforced by construction in your approach?
    3) why is this param init “convenient”?
* 104-106: the normal distribution (aka Gaussian distribution) is not the same as the uniform distribution; what exactly do you mean here?
* Figure 1: “Dashed links only exist if source network biases in layers l > 1 are initialized to zero.” -- isn’t it the converse, i.e., only when the biases are non-zero?
Assumption 2.2: why? what is the purpose of this type of block-diagonal symmetric structure that you impose on the source-net weight matrices?
* 136-150: this belongs in the related work section; in the following lines are several more repeated mentions of related work
* 153: explanation unclear. what is the middle layer?
* Direct citations should be made in the format Author et al. [1], (i.e. \citet in natbib not \citep). This formatting should be fixed throughout your work.

**Limitations:**

No such limitations/negative impacts are discussed by the authors. It's not immediately clear to the reviewer any potential impacts since this work is quite far from application, although pruning in general has been shown to have consequences for bias/robustness of trained models.

**Strengths And Weaknesses:**

Strengths:
* The authors attempt to reduce the depth provably required to find "Strong Lottery Tickets" i.e., subnetworks at initialization with good generalization.
* Any such guarantee could make a big difference in the feasibility of finding strong Lottery Tickets emperically.

Weaknesses
* The premise of the paper is entirely based on the (unstated) assumption that Lottery Tickets can approximate an arbitrary subnetwork's weights and structure (i.e. target network $f_t$, from source network $f_s$. This assumption is unproven at best, and in fact all evidence empirically points to this assumption likely being false. As has been shown empirically in multiple works, LTs are extremely restricted in the solutions they can find. In "Linear Mode Connectivity and the Lottery Ticket Hypothesis" by Frankle et al. show that all Lottery Ticket solutions are within the same basin, effectively learning very similar solutions. In "Gradient Flow in Sparse Neural Networks and How Lottery Tickets Win" by Evci et al, it is further shown that this solution basin is the same as the pruning solution, and a function similarity analysis shows they learn effectively the same function. Empirically we only know how to find one specific subnetwork with Lottery Tickets. Furthermore, we are not given any relationship between $f_t$ and $f_s$ (or any restrictions on $f_t$) to even understand how/if $f_s$ is related to the source model or the pruning solution.
* Given the small-scale model and dataset, the empirical results (on weak LTs, not strong LTs!) shown do not provide much support for the theoretical claim, notably the ReLU/LReLU results for depth L+1 are 0.2 percentage points lower than 2L and the target models. This might not seem significant, but when one considers how easy the problem is to solve, and how close the 2L solutions are, it is not as convincing. Please note these results are also Weak LTs, not strong LTs as the proof is intended to show the existence of.
* There are many other assumptions made critical to the proposed proof that are unjustified, not obvious, and in some cases obviously not maintained in real-world neural networks. For example, restricting the weight magnitudes of the model.
* Paper is very unclear in other sections, with math not well explained in bits, and language of some sections incomprehensible (more details below in Questions).

---

> ### Author Response · Authors · 2022-08-02
> **Answer to Reviever Puqv: Major Points of Critique**
>
> We are afraid Reviewer Puqv did not conceive the set-up and purpose of our existence proofs. We can only recommend reading carefully the section on background and notation in Section 2 and the section “Lottery Tickets as Subnetworks” that starts on page 3. These provide the basic information that is necessary to understand the overall objective. We otherwise do not comprehend how Reviewer Puqv could claim that our theory does not hold without pointing out any technical mistakes in the proof strategy. We also verify the validity of our explicit construction approach experimentally.
> By addressing the raised points of critique and correcting factually wrong statements by Reviewer Puqv, we hope to resolve most misunderstandings in the following.
>
> **Weakness 1: premise of the paper**
>
> We **prove** the existence of a LT that can approximate any given target network with the stated properties. That is **not an assumption but the **subject of our proofs**.
> Of course, the source $f_s$ is related to the target $f_t$. Note that the width of the source network needs to be wider by a logarithmic factor times the target width.
>
> To help Reviewer Puqv to clarify his/her confusion note that we do not assess the number of existing possibly different lottery tickets. The fact that LTs that are identified by specific pruning algorithms (e.g. IMP) are in the same basin does not challenge our results in any way. Actually, we want our LT to encode exactly the same function as the target network. Thus, the pointed out insights even highlight the relevance of our results on strong LTs, as the weak LTs are almost strong LTs if they do not change a lot during training. (If for some reason the reviewer wants to identify different LTs in experiments, a reasonable strategy to achieve that would be to change the task and thus the target network.)
>
> Also note that just because some pruning algorithms are not able to find LTs that solve a given task, this does not imply that no LTs or not target network exists. The reason for the failure could just be a limitation of the pruning algorithm or the fact that the source network was not wide enough to start with. According to the universal function approximation theorem, if the task can be solved with a continuous function, there exists a large enough neural network (the target) that can approximate that function and serve as our target network.
>
> **Weakness 2: Our empirical results provide a sanity check that our construction works.**
>
> We could easily reduce the overall error by increasing the approximation quality of each parameter by setting $\epsilon_l=0.001$ instead of $\epsilon_l=0.01$ if Reviewer Puqv thinks that this is relevant. We would also like to refer to our additional more large scale experiments in our answer for Reviewer NsjM, which provide additional experimental evidence.
> Note that our experiments construct **strong LTs** not weak LTs. We have identified suitable **target** networks with the help of IMP (if this is the source of confusion).
>
> We would further like to mention that explicitly pruning fully-connected networks for a target network is actually not such a trivial task because the associated matrices contain a high number of parameters and the approximation of each parameter involves the solution of an NP-hard subset sum approximation problem. This cannot be scaled up arbitrarily.
>
> **Weakness 3: Weight magnitudes can be restricted without loss of generality.**
>
> For that reason, it is quite standard to restrict weight magnitudes in LT existence proofs (see e.g. [35]). As explained in [3], we could simply adapt the subset sum approximation by multiplying the width requirement by the maximum absolute value of all target parameters. Our error propagation assessment does not even make the assumption on restricted weights.
>
> **Weakness 4: Clarity of writing.**
>
> We understand that Reviewer Puqv had clearly problems understanding the content of the paper (but is the only one with this issue). Our impression from the detailed questions is that he/she might also not have been open to the possibility that our theory is correct, which might be an explanation why he/she had a hard time.

---

> > ### Author Response · Authors · 2022-08-02
> > **Answers to detailed questions**
> >
> > To support a better understanding we try to answer the detailed questions by pointing out the locations where the answer can be found in our manuscript. However, there must have been a mix-up with some of the line numbers because sometimes the references make no sense.
> >
> > **26-34:** We cannot see any problem with this paragraph.
> >
> > **38-40:** How do these function approximation results relate to sparse nets? - The corresponding target networks would be very sparse and, consequently, also the constructed Lottery Tickets would be sparse. The point of this paragraph is that we explain the benefits of our theoretical advancements: The flexibility to use different activation functions and to leverage more depth of the source network could lead to potentially sparser target networks.
> >
> > **Target network:** Our results hold for any target network with the stated properties. In practice, this target network is inaccessible to us. This is why we try to find LTs instead. Note that even if we knew the target’s architecture, we might not be able to train the target network sufficiently with SGD. Just in our experimental sanity checks, we have taken trained neural networks as our targets.
> >
> > **67-69:** “Since the existence has been proven…” means that we refer to theoretical results and explain the general approach in this paragraph.
> >
> > **80:** Since  $\min\{\epsilon, \delta\}$ is part of our width requirement,  $\epsilon$ and $\delta$ can obviously be compared. We have only mentioned that practitioners usually care more about small $\epsilon$ rather than small $\delta$.
> >
> > **87-88:** See explanation above on weight bound.
> >
> > **95-98:**
> >
> > 1. The initialization is related to the activation function as in our theorems and as mentioned in the text. Our existence proofs for sigmoids (with $d \neq 0$) rely on Assumption 2.2, while the others rely on Assumption 2.1 in the 2layers for one construction.
> >
> > 2. A target layer is approximated by two source network layers according to our construction. We also explained the intuitive idea why we need that: We have to create multiple copies of neurons in the first layer to serve as basis for solving subset sum approximations of the target parameters.
> >
> > 3. As explained in lines 95-125, the convenient parameter initialization works directly for solving subset sum approximation problems. We could obtain the same result by rescaling a realistic parameter initialization that is used to make the source network trainable with SGD.
> > 104-106: We have just mentioned that our construction could be transferred to more general distributions (like Gaussian initializations) as explained in the appendix.
> >
> > **Figure 1:** The target biases can always be nonzero. If we initialize the biases in the source network to zero, however, we cannot use them for the approximation of the target biases. Instead we need to construct constant neurons with the help of the source weights (hence the dashed lines).
> >
> > The purpose of the looks-linear initialization is to tie the weights in such a way that we can completely remove their contribution on the nonzero intercept $d$ of the activation function (e.g. when we use sigmoids). See also the proof outline for Thm. 2.6.
> >
> > **153:** That’s a section header. No idea what the problem was. In general, the middle layers refer to any layer except the first and the last one.
> >
> >
> > **Assessment.**
> >
> > Considering that we have addressed all questions and Reviewer Puqv has not pointed out any factually wrong statements/proof elements/arguments or limitations of our work, we kindly ask Reviewer Puqv to increase his/her score taking this fact into account.

---

> > > ### Author Response · Authors · 2022-08-09
> > > **We would be happy about a response.**
> > >
> > > As the reviewer and author discussion period comes to an end, we kindly ask Reviewer Puqv for a response.
> > > As we believe that we have addressed all issues, we request that they reconsider their score. In case of remaining issues or questions, we would be happy to address them.

---

> > ### Comment · Reviewer_Puqv · 2022-08-09
> > **RE: Major Points of Critique**
> >
> > While I appreciate the passion of the authors for their own work, I did read the paper, and suggesting I misunderstood everything and should just re-read it isn't a helpful comment in rebuttal. Frankly your "Weakness 4" rebuttal on the clarity of writing is particularly unprofessional. I'm always quite open to considering I misunderstood something, but the authors should also consider that they have the responsibility of writing a paper that is accessible and understandable to their audience. Please consider that the average reader doesn't ever put as much time into reading a paper and understanding it as your reviewers will.
> >
> > *Constructive feedback for your future rebuttals*: avoid blaming the reviewer, instead think about how you could make what's misunderstood more clear. It's also just a really bad way of motivating someone, who has volunteered to spend their evening reading your paper and rebuttal on openreview instead of with their family, from further engaging with you/trying to understand the paper better.
> >
> > With that out of the way, here I will go through each of my major points and discuss if you have addressed these in your rebuttal.
> >
> > ### Weakness 1
> >
> >  >Actually, we want our LT to encode exactly the same function as the target network
> >
> > This is exactly my point. As far as we know, LTs can't approximate specific functions arbitrarily, and yet this appears to be a requirement of your proof as currently stated, in part because you don't define (or even tell us in the rebuttal) the relationship between the target model and source functions. In the general case it doesn't appear it is empirically at least, and nobody has proved this for arbitrary functions.
> >
> > > Of course, the source fs is related to the target
> >
> > My question was specifically how are they related, not if they are related. As you will be aware this is fundamental in the LTH setting. As-is you have not clarified the issues with the definition and lack of relationship between the source and the target.
> >
> > > Thus, the pointed out insights even highlight the relevance of our results on strong LTs, as the weak LTs are almost strong LTs if they do not change a lot during training
> >
> > I find this to be a problematic statement that seems to highlight a disregard to the differences between strong and weak LTs in the paper in general, or perhaps a fundamental lack of understanding on the difference. Strong LTs are trivially a weak LT, but this does not hold the other way around.
> >
> > Strong LTs are LTs found at random initialization with no training. To be clear weak LTs are found by: (i) training a dense model to good generalization, (2) pruning it (sometimes iteratively), (iii) weight re-winding to find weights (i.e. the weak LT) that can be trained to good generalization. There is no almost strong LT, it's trained or it isn't.
> >
> > ### Weakness 2
> > I would like to understand what is relevant about this experiment, if not that you are trying to show you can approximate the target function well with the LTs found? Frankly I think good results on MNIST/LeNet are not a strong ask. However, there are other significant issues with this experiment aside from the poor results on MNIST with LeNet we can discuss:
> >
> > (a) Your statement in the rebuttal that your experiments use strong LTs appears to be contradicted by the text, Section 3, "As the influential work by [13], we use Iterative Magnitude Pruning (IMP) on LeNet networks 378 with architecture [784, 300, 100, 10] to find LTs". This is by definition a weak LT, and not a strong LT. This seems like a massive issue if your proof is about showing the existence of strong LTs?
> >
> > (b) Evaluating the generalization alone is not an evaluation of the function similarity. Two models can have very similar generalization for example on a classification task, and yet very different predictions on individual samples. It has been shown that LTs learn very similar functions to the original pruned function they are based on (and each other).
> >
> > ### Weakness 3
> >
> > If this has been used in [35]/explained in [3] then reference these in the statement, saying WLOG requires it to be trivially obvious or explained.
> >
> > ### Weakness 4
> >
> > Apparently we'll just have to agree to disagree on this one.

---

> > > ### Author Response · Authors · 2022-08-10
> > > **We appreciate the answer of Reviewer Puqv**
> > >
> > > **Feedback on rebuttal**
> > >
> > > We appreciate the feedback by Reviewer Puqv and the time spent on reviewing.
> > > We did not mean to hurt their feelings with the statement that they misunderstood the paper but we had to point it out to clarify the problem.
> > > In fact, we answered every raised question, also the ones that were just a matter of taste, and tried to facilitate a better understanding. We also believe that pointing out the sections where we have provided the information that Reviewer Puqv is looking for is constructive.
> > >
> > > Unfortunately, we are aware that asking for more openness when reading our paper is probably not resulting in more openness but we did not know how we could have achieved that otherwise.
> > > The fact that Reviewer Puqv chose to only answer us past the deadline without giving us the ability to respond actually supports our point. It is certainly not constructive and suggests that Reviewer Puqv is not interested in or open for an answer.
> > >
> > > Of course, we wonder why Reviewer Puqv gave us a 2 rating that is frankly inappropriate for a paper that is not technically wrong.
> > >
> > > Regardless, we would be happy to try again by addressing the raised points and clarify further misunderstandings below.
> > >
> > > **Weakness 1**
> > >
> > > Note that, as we have proven, LT pruning can in theory approximate any target network if the source networks is wide enough relative to the target network. (Btw, this has also been proven before for ReLU networks, in case that Reviewer Puqv prefers those results.)
> > > The fact that LT pruning algorithms fail sometimes makes such a theoretical insight even more important. It tells us that a) our pruning algorithms might succeed if we make the source network wider or b) that our contemporary pruning algorithms are not good enough (yet) to find existing lottery tickets.
> > >
> > > Note that the source network is a randomly initialized network that is simply wider than all target networks that could be approximated by pruning the source network. The relationship between the source and the target network is really just defined by comparing their width and depth, as we also explained in our answer. All relationships are stated precisely in our theorems.
> > >
> > > Maybe Reviewer Puqv is in fact more interested in the relationship between the lottery ticket and the target network? The lottery ticket approximates a given target network and is explicitly constructed in our proofs. The lottery ticket encodes almost the same function as the target network (up to error $\epsilon$).
> > >
> > > Note that we are very well aware how weak lottery tickets are defined and obtained. We just do not understand, how the specific pruning algorithms are relevant to our discussion and what exactly Reviewer Puqv perceives as Weakness 1 now.
> > >
> > > **Weakness 2**
> > > (a) The purpose of our experiments is to verify our theory. As our proofs are constructive and obtain lottery tickets by pruning a source network to approximate a target network, we follow the steps of our proofs to construct strong lottery tickets. Note that this means that we need two input networks: a target network and a random source network. By pruning the random source network we obtain a lottery ticket (the output of the algorithm) so that is approximates the target network.
> > > As explained in our previous answer, the weak LeNet LTs that we obtained by IMP serve es our **target** networks. Based on these targets, we construct **strong** lottery tickets.
> > >
> > > Also note that we extended our experiments considerably following the great suggestions of Reviewer NsjM.
> > >
> > > (b) We just followed the definition of a lottery ticket. The only requirement in the definition is (and the relevant aim is) that it achieves a similar or better performance in comparison with the trained complete network or, in our case, the target network (since this is already an accepted lottery ticket).
> > >
> > > Since the lottery ticket approximates the target network, it also encodes a similar function, but this aspect is not relevant for LT existence.
> > >
> > > **Weakness 3**
> > > We would be happy to add the two references as an explanation. We thank Reviewer Puqv for the suggestion.

---

### Official Review · Reviewer_MTH7 · 2022-07-12

**Rating:** 5
**Confidence:** 5
**Soundness:** 4 excellent
**Presentation:** 4 excellent
**Contribution:** 2 fair

**Summary:**

This paper studies the Strong Lottery Ticket Hypothesis (SLTH) for non-ReLU activation functions, just as its title implies. Previous theoretical works have advanced the SLTH in many aspects. One missing part is that their theories focus on the ReLU case, while this paper extends them to other common activation functions and maintains the L+1 depth and logarithmic width requirements. They mainly focus on FC networks in this work. Empirical study on MNIST is conducted to validate their theory.


**Questions:**

* Please respond to my concerns above
* Side question: SLTH is interesting itself. However, as the experiments in this paper imply, we need much more parameters to realize it. I am wondering, what possible *practical* meaning it could imply?


**Limitations:**

Yes, they address the limitations and potential negative societal impact of their work.


**Strengths And Weaknesses:**

**Pros**
1. They extend SLTH to non-ReLU activation functions, which is a credible theoretical contribution towards LTH, network pruning, or neural networks themselves.
2. Meanwhile, the L+1 depth and logarithmic width requirements are maintained.
3. The paper is well-written and well presented.

**Cons**
I like this paper on its own. My major concern is that, given the existence of [1] (L71: “While this approach has been transferred to convolutional and residual architectures [1]), what is *truly new* from this work?

The major construction scheme in this paper (Sec. 2) has a non-trivial overlap with [1] (Sec. 2 and Sec.3). The major novelty of the non-ReLU activation functions for SLTH also has been discussed in [1].

For the rating - I think this is a good paper in general. But unfortunately, some major content has appeared in previous works. My final rating hinges on how the authors respond to my concerns.

---

> ### Author Response · Authors · 2022-08-02
> **Our work is the first to prove the LTH for activation functions different from ReLUs for fully connected networks.**
>
> We thank Reviewer MTH7 for the generally positive assessment of our work and address the two raised questions as follows.
>
> 1. Regarding the overlap with [1], we would like to point out that [1] itself refers to the submitted paper. As mentioned on page 1, page 2, page 4, and on page 6 of [1], the main idea to approximate a given activation function by a leaky ReLU is developed in the present paper and also explained in detail here. [1] is concerned with the transfer to convolutional and residual structures, which are quite different from fully connected networks. This entails different error propagation assessments, different approximations of single neurons, a different definition of g and the scaling in the 2-layer construction, even different parameter initializations are required. Note that [1] does not even discuss activation functions with $d \neq 0$ (like sigmoids) but refers to our paper instead. To be fair, [1] would need to do some extra work to derive the right initialization that would apply to the context of convolutional and residual layers.Thus, the transfer is not immediately clear but probably possible.
>
> 2. Reviewer MTH7 raises a very interesting question regarding the practicality of the SLTH. It is correct that at least in theory the source network and the LT consist of more parameters than the actual target network. (However, if the source network is sufficiently large, the LT might not be significantly larger than the target network.) It is an open question whether strong LTs have a disadvantage in comparison with weak LTs that can be trained after pruning. The reason is that we have not really understood the degree of overparametrization that is required for training the weak LT successfully with SGD. At least experimentally, contemporary pruning algorithms do not seem to be always able to identify LTs that are as sparse as a potential target network (see e.g. [12]). In general, it seems to help if the weak LT is initially already close to the final trained state (which gives it similar properties as a strong LT). For that reason, in the process to identify extremely sparse neural network architectures, it might be necessary to not restrict ourselves to pruning alone but allow for additional processing steps that could be different from training with SGD or for combining ideas pertaining to strong LTs, weak LTs, and general compression approachs. There are some recent exciting developments in this direction. For instance, we can highly recommend: [https://arxiv.org/pdf/2202.12002.pdf](https://arxiv.org/pdf/2202.12002.pdf),  [https://arxiv.org/pdf/2203.04248v1.pdf](https://arxiv.org/pdf/2203.04248v1.pdf), or [https://proceedings.mlr.press/v162/chen22a.html](https://proceedings.mlr.press/v162/chen22a.html).

---

> > ### Comment · Reviewer_MTH7 · 2022-08-07
> > **Thanks for the clarification**
> >
> > > [1] is concerned with the transfer to convolutional and residual structures, which are quite different from fully connected networks.
> >
> > I am still concerned about whether the difference between CNN and FC networks is significant enough to differentiate this paper from [1]. As is known, if the kernel size is 1x1, a CNN would turn into an FC network. That is, **we can consider FC networks as a special case of CNNs**. Since [1] has covered the (major) topic of this paper in the case of CNN, it seems the FC case has already been done by extending the conclusion in [1] to 1x1 kernels.
> >
> > > This entails different error propagation assessments, different approximations of single neurons, a different definition of g and the scaling in the 2-layer construction, even different parameter initializations are required.
> >
> > What are the differences specifically in the paper? Especially, how would the kernel size shrinking from like 3x3 to 1x1 make a substantial difference in your theory?

---

> > > ### Author Response · Authors · 2022-08-08
> > > **[1] is based on this paper.**
> > >
> > > We thank Reviewer MTH7 for acknowledging our response and giving us the opportunity to develop our argument in more detail.
> > >
> > > We would like to stress that the $L+1$-construction and the extension to activation functions different from ReLUs is not a contribution of [1]. It references this paper for a detailed exposition. We argue that this is an argument in favor of our work, as this highlight the relevance of the presented results.
> > >
> > > It is correct that fully-connected networks could be seen as special case of convolutional architectures with 1x1 filters. However, the general proof and construction idea is explained here. Just additional complications that are specific to convolutional and residual architectures are discussed in [1].
> > >
> > > 1. As a consequence, [1] does not cover activation functions with nonzero intercept ($d \neq 0$) that work together with a looks-linear initialization. Thus, Theorem 2.6 has not been transferred to convolutional architectures in [1] and the proof for sigmoids (and other activation functions with $d \neq 0$) is unique to this paper.
> > >
> > > 2. Another consequence is that the error assessment is stronger in this paper.
> > > Our allowed error in Layer l is $\epsilon_l \propto  \frac{\epsilon}{n_{l} L} \left(1+\frac{\epsilon}{L}\right)^{l-L} $. This is much better (larger allowed error) than the error that we would get from using [1], which is divided by a potentially large sum over all nonzero target parameters and a product over nonzero parameters in the upper layers: $\epsilon_l \propto  \frac{\epsilon}{ (\prod^L_{s=0} N_{t,s} )\prod^L_{s=l} N_{t,s}} $.
> > >
> > > 3. This error also enters the function $g^{-1}$ (see Eq. (9) here and (12) in [1]) and thus determines the scaling of the initial parameters. Thus, parameters in [1] would need to be initialized as much smaller in the first layer than in our case. From a practical point of view this might mean that our construction is numerically more stable if the source network is very large. From a theoretical perspective, however, the dependence of the width requirement on the parameters of the target network is the most interesting difference as discussed in 2.

---

### Author Response · Authors · 2022-08-02
**Summary of Response to Reviewers**

We thank all reviewers for the constructive criticism and have revised our manuscript accordingly:

1. Most importantly, we have added new larger scale experiments based on ResNet18 networks and tiny-ImageNet.

2. Additionally, in a concrete example, we have highlighted potential benefits of a $L+1$-construction in comparison with a $2L$-construction that originates in the representational benefits that are available to a target network of higher depth.

We are confident that our work provides novel theoretical insights of interest to the NeurIPS community, as we have extended the strong lottery ticket hypothesis to non-ReLU activation functions and target networks that have almost the same depth as the random source network, which is pruned to obtain lottery tickets. These innovations increase our flexibility to represent potentially sparser target networks and, in consequence, provide evidence that it might be possible to prune narrow source networks to obtain sparse lottery tickets.
Our results are based on several technical innovations regarding the explicit construction of lottery tickets based on a given target network.

---

### Meta-Review · Area_Chair_nVpE · 2022-08-23

**Recommendation:** Accept
**Confidence:** Certain

**Metareview:**

The submission provides a nice extension of the previous work that shows a random network with 2L layer contains a subnetwork (or lottery ticket) that approximates the target network of depth L well. Instead of 2L, they show L+1 layer plus a logarithmic factor wide suffices. Overall it is a nice solid contribution to the community. In the camera ready version, AC would advise the authors to clearly specify the difference between this submission and [1].

The concerns raised by reviewer Puqv are due to misunderstanding of lottery ticket hypothesis (LTH) and are largely not valid. For example, "Linear Mode Connectivity and the Lottery Ticket Hypothesis" by Frankle et al, in fact shows *after the initial phase of the network training*, all Lottery Ticket solutions are within the same basin. For different randomly initialized networks, the resulting LTHs can be very different.

**Award:**

No

---

### Decision · Program_Chairs · 2022-09-14

Accept